# Development of a new largely scalable *in vitro* prion propagation method for the production of infectious recombinant prions for high resolution structural studies

Hasier Eraña[1,2], Jorge M. Charco[1], Michele A. Di Bari[3], Carlos M. Díaz-Domínguez[1], Rafael López-Moreno[1], Enric Vidal[4], Ezequiel González-Miranda[1], Miguel A. Pérez-Castro[1], Sandra García-Martínez[1,2], Susana Bravo[5], Natalia Fernández-Borges[1], Mariví Geijo[6], Claudia D'Agostino[3], Joseba Garrido[6], Jifeng Bian[7], Anna König[8,9], Boran Uluca-Yazgi[8,9], Raimon Sabate[10,11], Vadim Khaychuk[7], Ilaria Vanni[3], Glenn C. Telling[7], Henrike Heise[8,9], Romolo Nonno[3], Jesús R. Requena[12], Joaquín Castilla[1,13]*

1 CIC bioGUNE, Derio (Bizkaia), Spain, 2 ATLAS Molecular Pharma S. L. Derio (Bizkaia), Spain, 3 Department of Veterinary Public Health and Food Safety, Istituto Superiore di Sanità, Rome, Italy, 4 Centre de Recerca en Sanitat Animal (CReSA), UAB-IRTA, Barcelona, Spain, 5 Proteomics Lab, IDIS, Santiago de Compostela, Spain, 6 Animal Health Department, NEIKER-Instituto Vasco de Investigación y Desarrollo Agrario, Derio (Bizkaia), Spain, 7 Prion Research Center (PRC), Colorado State University, Fort Collins, Colorado, United States of America, 8 Institute of Complex Systems (ICS-6) and Jülich Center for Structural Biology (JuStruct), Forschungszentrum Jülich, Jülich, Germany, 9 Physikalische Biologie, Heinrich-Heine-Universität Düsseldorf, Düsseldorf, Germany, 10 Department of Pharmacy and Pharmaceutical Technology and Physical-Chemistry, Faculty of Pharmacy and Food Sciences, University of Barcelona, Spain, 11 Institute of Nanoscience and Nanotechnology (IN2UB), University of Barcelona, Spain, 12 CIMUS Biomedical Research Institute, University of Santiago de Compostela-IDIS, Spain, 13 IKERBasque, Basque Foundation for Science, Bilbao (Bizkaia), Spain

* castilla@joaquincastilla.com

**Data Availability Statement:** All relevant data are within the manuscript and its Supporting Information files.

## Abstract

The resolution of the three-dimensional structure of infectious prions at the atomic level is pivotal to understand the pathobiology of Transmissible Spongiform Encephalopathies (TSE), but has been long hindered due to certain particularities of these proteinaceous pathogens. Difficulties related to their purification from brain homogenates of disease-affected animals were resolved almost a decade ago by the development of *in vitro* recombinant prion propagation systems giving rise to highly infectious recombinant prions. However, lack of knowledge about the molecular mechanisms of the misfolding event and the complexity of systems such as the Protein Misfolding Cyclic Amplification (PMCA), have limited generating the large amounts of homogeneous recombinant prion preparations required for high-resolution techniques such as solid state Nuclear Magnetic Resonance (ssNMR) imaging. Herein, we present a novel recombinant prion propagation system based on PMCA that substitutes sonication with shaking thereby allowing the production of unprecedented amounts of multi-labeled, infectious recombinant prions. The use of specific cofactors, such as dextran sulfate, limit the structural heterogeneity of the *in vitro* propagated prions and makes possible, for the first time, the generation of infectious and likely homogeneous samples in sufficient quantities for studies with high-resolution structural techniques as

**Funding:** This work was supported financially by several Spanish grants awarded to JC (AGL2015-65046-C2-1-R and BFU2013-48436-C2-1-P) and JRR (BFU2017-86692-P) by MINECO/FEDER, as well as an Interreg (POCTEFA EFA148/16) grant awarded to JC by FEDER. The funders had no role in study design, data collection and analysis, decision to publish, or preparation of the manuscript.

**Competing interests:** I have read the journal's policy and have the following conflicts: Authors declare that Hasier Eraña and Sandra García-Martínez are employed by the commercial company ATLAS Molecular Pharma SL. This does not alter our adherence to all PLOS Pathogens policies on sharing data and materials.

demonstrated by the preliminary ssNMR spectrum presented here. Overall, we consider that this new method named Protein Misfolding Shaking Amplification (PMSA), opens new avenues to finally elucidate the three-dimensional structure of infectious prions.

## Author summary

Prion disorders are a group of devastating neurodegenerative diseases caused by an aberrantly folded isoform of the endogenous prion protein. The molecular mechanisms by which this proteinaceous pathogen is able to propagate in the central nervous system and cause neuronal death are poorly understood, partially due to the difficulties elucidating the three-dimensional structure of the aggregation-prone aberrant isoform or prion. Obtaining sufficient amounts of highly pure and homogeneous prions for high-resolution structural studies has been limited until now due to technical reasons. Here, we present a novel method for the production of large amounts of highly infectious recombinant prions suitable for solid state Nuclear Magnetic Resonance imaging, which could help to unveil the molecular pathogenesis of these particular pathogens.

## Introduction

The generation of recombinant prions *in vitro* able to cause Transmissible Spongiform Encephalopathy (TSE) *in vivo* has been one of the greatest advances of the last decades in the field [1–12] and has been successfully achieved by several research groups, although there are still several unsolved issues that make this process of particular interest: 1) The recombinant prions generated by different research groups applying similar techniques yielded highly variable results. From generating prions with infectivity similar to that of some mammalian prions [8, 10], to producing prions with very low infectious ability or even lacking infectivity *in vivo*, requiring in some cases multiple *in vivo* passages to cause clinical prion disease [1–3, 6]. 2) It is unknown whether any other component, besides PrP$^{Sc}$, may be the infectious particle. Many different molecules have been tested (RNA, lipids, dextran sulfate, plasmid, etc.) and, in some cases, those are claimed as a requirement for *in vitro* infectious prion formation [4, 8, 13–15], however, credible evidence is lacking. There are several methods capable of generating misfolded self-propagating recombinant PrP but given the disparity of infective abilities, it is impossible to clarify which is the best to generate infectious recombinant prions and which may yield only non-infectious misfolded recombinant PrPs. 3) Finally, these techniques could help to determine the critical difference between infectious and non-infectious self-propagating, protease-resistant misfolded recombinant PrPs.

The vast majority of studies for *in vitro* recombinant prion generation have been performed using PrPs from well-known rodent species, mainly mouse (*Mus musculus*) [1, 8] and hamster (*Mesocricetus auratus*) [5]. Both are some of the best-characterized models of TSE and are the preferred ones for evaluating *in vivo* infectivity due to short incubation period and ease of handling. Recently, another rodent species, bank vole (*Myodes glareolus*) has become increasingly popular due to its particular attributes. A polymorphic variant of the bank vole PrP, bearing isoleucine at position 109, shows the ability to misfold spontaneously *in vitro* [10] and its overexpression in transgenic mice leads to a spontaneous and transmissible prion disease [16]. The bank vole is also considered to be an almost universal acceptor of prions since can be infected with a large diversity of prion strains from different donor species [17] and when infected with

Chronic Wasting Disease (CWD), a TSE from cervids, has the shortest incubation period of any known prion disease [18].

This combination of bank vole-based rodent models and the development of new generation recombinant prion propagation systems may finally decipher the mechanisms of infectivity of prions and may allow elucidation of the different structural features of various prion strains. The generation of the first highly infectious recombinant prion, using recombinant mouse PrP complemented with RNA and lipids, resulted in spontaneous misfolding of the protein using protein misfolding cyclic amplification (PMCA) [8]. This proved that prions able to cause clinical disease *in vivo* could be generated *in vitro* with minimal synthetic components. The presence of lipids and RNA in the mixture and previous research claiming the need of cofactors for the generation of infectious recombinant prions [12, 14, 15, 19–21] suggested that recombinant PrP alone was not sufficient to give rise to infectious prions *in vitro*. However, the actual role of cofactors was never investigated further so the true characteristics of prion infectivity remain unknown.

Recently, infectious recombinant prions have been generated in the absence of cofactors using PMCA and recombinant PrPs as the reaction substrate [10]. Recombinant bank vole I109 prions were generated by two distinct procedures: spontaneously and using a seed based on infectious prions of mammalian origin [18]. Several distinct infectious recombinant prion strains were generated in both ways using a variety of substrates containing different cofactors: from complete brain homogenates of PrP knock-out transgenic mice (*Prnp*$^{0/0}$) to substrates containing cofactors such as RNA, dextran sulfate or plasmid DNA, or in the absence of any cofactor. Among the conclusions derived from this research is that, in a cofactor rich environment (i.e. PrP and *Prnp*$^{0/0}$ brain homogenate), the recombinant PrP misfolds in many different conformations, giving rise to a mixture of strains with distinct properties regarding their electrophoretic migration patterns and their abilities to cause disease *in vivo*. This is in agreement with recent studies on prion strain evolution that propose the existence of pools of slightly different prion conformers called prion substrains or quasispecies [22–24]. Our previous work, in which substrates with minimal components (i.e. dextran sulfate, RNA and plasmid DNA) were proven useful to select or specifically propagate certain strains [10], highlighted the possibility of generating more homogeneous samples that could facilitate the application of ssNMR to infectious prions. Although the selection of a single strain could not be definitively demonstrated, our results clearly indicated that using selective substrates, the conformer variety of the original seed could be reduced.

To understand the nature of prion infectivity, which would allow us to understand the nature of different prion strains, knowledge of the three dimensional structure of different prions at the atomic level appears necessary [25]. Techniques such as Hydrogen/Deuterium exchange [26–28], limited proteolysis coupled to mass spectrometry [29–33] and electron microscopy [34, 35] have been used for that purpose and the information obtained from them was highly valuable although insufficient to definitively understand the essence of prion infectivity. The incomplete knowledge about prion structural features is obvious considering the radically distinct structural models currently proposed for prions [34, 36, 37], both based on different techniques. The limited success of the aforementioned techniques and the constraints imposed by infectious prions, which are intrinsically aggregated making them unsuitable for high resolution techniques such as X-ray crystallography or nuclear magnetic resonance (NMR) imaging, reduces the possibilities of determining the atomic structure of infectious prions except by techniques such as solid state NMR (ssNMR) [38]. This technique has already resolved the structures of proteins with similar properties to mammalian prions like HET-S [39], Aβ amyloid [40] and even a synthetic infectious PrP23-144 amyloid that seems to propagate *in vivo* by templating endogenous PrP$^C$ into PrP$^{Sc}$ [41]. Despite its potential, ssNMR has

only been successfully applied to the resolution of a single truncated mammalian prion structure [42] due to two main limitations as it requires: 1) significant homogeneity of the sample under study and, 2) large amounts of isotopically labeled material.

Here, we present a new technique for propagation *in vitro* of infectious recombinant prions that overcomes the limitation of generating large amounts of isotopically labeled sample required for ssNMR studies. The procedure is based on PMCA, from where the misfolded recombinant seed is obtained. This infectious seed has been adapted to a new system in which shaking [43] is used instead of sonication for *in vitro* propagation of recombinant prions and this novel propagation method has been named Protein Misfolding Shaking Amplification (PMSA). This methodology eliminates the need of complex and expensive equipment permitting its implementation in any basic laboratory and allows the generation of 2.5 mg of protease-K digested infectious recombinant prion per day ready to be studied. The optimization applied to PMSA permits greater than 1,000-fold scaling up from minute amounts of seed of isotopically labeled material (and also multiple labeling) suitable for ssNMR studies, as shown by the preliminary spectrum presented here.

## Results

### Selection of an infectious recombinant prion for ease of indefinite *in vitro* propagation

With ssNMR studies in mind, a recombinant misfolded PrP was selected that fulfilled the following ideal characteristics: 1) minimal additional components required to be propagated *in vitro*, 2) additional components cheap and easily obtainable, 3) highly permissible to *in vitro* propagation allowing rapid generation of large amounts, and 4) infectious *in vivo* (100% attack rate with reasonable incubation times using a wild-type or no overexpression model).

In a previous study, a recombinant bank vole PrP complemented with $Prnp^{0/0}$ mouse brain homogenate (to provide any cofactor from the brain that may be required) was misfolded by PMCA (named H/L-seeded-03). The resultant strain mixture could be selected or adapted to different *in vitro* propagation environments in which the $Prnp^{0/0}$ brain homogenate was replaced by single cofactor complemented substrate such as dextran sulfate or RNA [10]. The latter recombinant seeds, requiring minimal additional components, were infectious *in vivo*, resulted in classical electrophoretic migration patterns and were more homogeneous than those complemented with brain homogenate, in which strain mixtures were detected upon *in vivo* inoculation [10]. Among these, the dextran-complemented recombinant prion (from now on named L-seeded-PMCA) resulted in slightly higher attack rates and much shorter incubation times, as well as a better *in vitro* propagation capacity [10]. Therefore, the recombinant bank vole seed generated in $Prnp^{0/0}$ brain homogenate and adapted to dextran-complemented substrate by serial PMCA rounds was chosen as the most suitable prion to adapt to PMSA.

### Adaptation of the chosen recombinant prion to the new PMSA method

PMSA was developed based on PMCA, but seeking increased product output. Its success lies on propagating at least 1,000 times larger volumes of misfolded recombinant PrP, from the hundreds of microliters that can be propagated by PMCA (space limited by plate sonicators) to hundreds of millilitres, avoiding the use of complex or expensive equipment.

Therefore, the first step consisted on designing a method avoiding sonication, but efficient enough propagating the seed of choice. To address this, all the parameters used in PMCA for recombinant prion propagation (i. e. temperature, incubation times, number of propagation rounds) were retained except the quantity limiting sonication, which was substituted with

shaking. Standard shakers with appropriate temperature control would allow development of a highly scalable method by designing specific racks for tubes of any size allowing the production of the volumes required for structural studies. The L-seeded-PMCA recombinant seed, of proven infectivity and showing distinctive strain features [10], was propagated serially for 40 rounds at 1:10 dilution over a substrate containing bank vole recombinant PrP with dextran sulfate in order to adapt it to PMSA (named L-seeded-PMSA). Its stability through the serial rounds was checked and compared to the PMCA-derived prion also. Both the electrophoretic patterns (Fig 1A and 1B), as well as the abilities to propagate by PMCA in whole brain homogenate substrates (indicator of their potential *in vivo* infectivity), were conserved suggesting adaptation to PMSA (Fig 1C).

Once the seed was adapted to serial propagation in the novel system, the propagative abilities of both methods were compared by performing a serial dilution of each seed over a single round of PMCA or PMSA (Fig 2). Both seeds showed a similar propagation capacity after applying both methodologies. Dilutions of $10^4$–$10^5$ could be detected by WB after PK treatment, a reasonable dilution but suggesting the need for optimization.

## Optimization

To improve the propagative capacity of the PMSA, all parameters possibly relevant for PrP misfolding were considered and the propagation ability evaluated by serial dilution of the seed in a variety of conditions. Optimisation of the parameters for recombinant prion propagation was performed using three different shakers. Two different shakers with thermoblocks (Thermomixer from Eppendorf and a digital heating shaking drybath from Thermo-Scientific) and a simple plate shaker without temperature control placed in an oven to control temperature. The outcome of each parameter-variation was evaluated by Thioflavin T measurement and by proteinase K digestion and Western blotting as detailed in Materials and methods. The initial parameters set for PMSA were an adaptation of our regular recombinant PMCA settings. Recombinant protein concentration (2 μM) and PMCA substrate preparation were conserved as well as the temperature (38˚C), addition of 1 mm zirconia/silica beads and 24 h propagation rounds. As the energy provided by the system is presumably lower than by sonication, the maximum speed of the shakers (1,200 rpm) was used and to increase shaking cycles with respect to PMCA, cycles of 60 s shaking and 5 min incubation were used. However, all parameters were modified to optimize the propagation capacity evaluated by serial dilution of the recombinant seed in a range of different parameters. Shaking speed, shaking/incubation cycles, type of beads, temperature and rec-PrP concentration were optimized. Optimal conditions were found to be shaking at 1,000 rpm continuously, with 1.0 mm zirconium silicate beads at 39˚C and 2 μM of rec-PrP (Fig 3).

## Scalability of PMSA

Once the propagation parameters were optimized, the scalability and repeatability of the novel method needed to be assessed. A serial dilution study was performed using tubes of different size with specific adaptors, designed by 3D printing, for each of them (Supplementary information, S1 Fig). PCR tubes (volume 50 μl), 2 ml tubes (volume 1 ml) and 5 ml tubes (volume 4.5 ml) were tested, as the scalability levels demanded to the system required tubes where large volumes could be propagated. In all the cases, 1 mm zirconium silicate beads were added and the tube orientated to favour bead movement. Propagation capacities were compared by serial dilutions of the seed as described previously.

Propagation was most and equally efficient in the 2 and 5 ml tubes, although all tube sizes resulted in reasonable propagative capacity. Thus, 2 and 5 ml tubes, apart from being the best

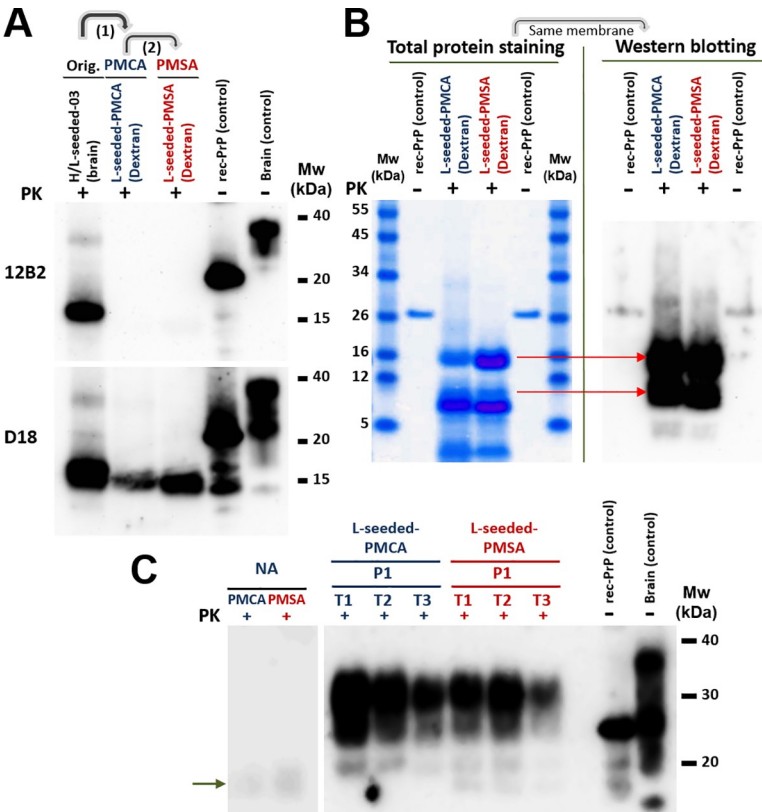

**Fig 1. Adaptation of the infectious recombinant prion (L-seeded-PMCA) to the new *in vitro* propagation method (L-seeded-PMSA). A)** Immunodetection of the characteristic protease resistant core resulting from cleavage of the amino-terminal tail of the original PMCA-derived recombinant prions (H/L-seeded-03 and L-seeded-PMCA) and the PMSA-adapted one (L-seeded-PMSA) by Western blotting. Samples were digested with 85 µg/ml of proteinase-K (PK) and analyzed by Western blotting using monoclonal antibodies 12B2 (1:2,500) and D18 (1:5,000). Differential digestion of the N-terminal makes the two recombinant prions adapted to propagate in dextran sulfate-complemented substrates (L-seeded-PMCA and L-seeded-PMSA) undetectable by 12B2 mAb (epitope 88–92) whereas the $Prnp^{0/0}$-complemented sample (H/L-seeded-03), which according to previous results (10) is a mixture of strains, detectable by both 12B2 and D18 mAbs (epitope 143–149). **B)** Electrophoretic migration patterns of L-seeded-PMCA and L-seeded-PMSA recombinant prions. The two recombinant prions were digested with PK and analyzed by BlueSafe staining (total protein) to examine all the proteolytic fragments derived from digestion. The same gel was transferred for Western blotting and developed with D18 mAb. Both recombinant misfolded PrPs show a similar banding pattern, suggesting similar if not identical conformations. The differences on electrophoretic patterns detected by D18 antibody between the gel in panel A and the one in panel B are due to the amount of sample loaded in each case. For total protein staining and posterior western blotting from panel B, 50x more sample is loaded than for panel A, making some minor PrP fragments detectable. **C)** Determination of potential infectivity (ability to propagate on brain-derived PrP) of L-seeded-PMCA and L-seeded-PMSA recombinant prions by brain-PMCA. Conservation of the capacity to misfold brain-derived PrP$^C$, as an indication of potential *in vivo* infectivity, was evaluated using equal amounts of the two recombinant inocula as seeds in TgVole brain homogenate at 1:10 dilution during one round of PMCA. Triplicates (T1, T2 & T3) were performed for both samples and PK-resistant PrP formation was monitored by Western blotting with D18 mAb. Misfolded brain-derived PrP arises from the first round for the dextran sulfate-complemented seeds with no difference between PMCA- or PMSA-adapted seeds (L-seeded-PMCA and L-seeded-PMSA). (→) Signals of PK-digested misfolded rec-PrPs appear lower than those of the non-glycosylated band due to the absence of the GPI anchor. rec-PrP (control): undigested bank vole rec-PrP protein. Brain (control): undigested TgVole whole brain homogenate. NA: Non-amplified samples.

vessels to reach the amounts of misfolded protein required ([Fig 4]) are easier to handle than the smaller ones and with just one standard shaker and oven allowed the generation of milligrams of misfolded sample per day (with 10 shakers, up to 400 ml can be produced daily).

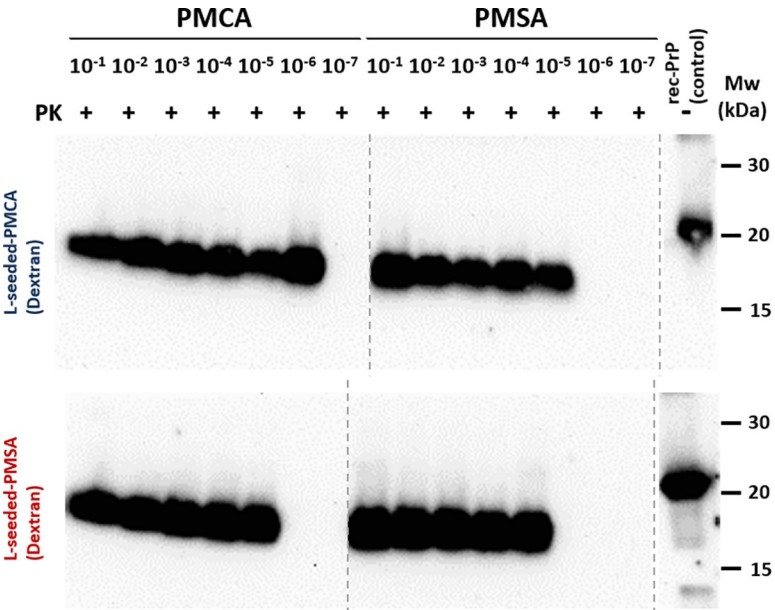

**Fig 2. Propagation capacity of L-seeded-PMCA and L-seeded-PMSA recombinant prions in PMCA and PMSA.**
The propagation efficiency of both recombinant prions *in vitro* in PMCA in comparison with PMSA was evaluated by serial dilutions of the seeds (from $10^{-1}$ to $10^{-7}$) in the same substrate (bank vole 109I rec-PrP complemented with dextran sulfate) subjected to 24 h rounds of PMCA and PMSA. Protease resistant rec-PrP formation was monitored by PK-digestion and western blot, using D18 mAb (1:5,000). L-seeded-PMSA showed a similar propagation capacity to PMCA. This blot shows a slight difference (1 log) when comparing the propagation capacity of both seeds in PMCA but not in PMSA. However, these small differences were not significant. Despite all the samples being run at the same time, the blot was cropped as indicated by the vertical dotted line to avoid displaying unrelated samples. rec-PrP (control): undigested bank vole rec-PrP protein.

### *In vitro* propagation efficiency

To estimate the efficiency of prion propagation by PMSA, the formation of protein aggregates that could be pelleted by centrifugation was undertaken. Since infectious PK-sensitive aggregates have also been described [31], the quantification of only protease-resistant misfolded PrP was considered inaccurate. However, the formation of amorphous aggregates without infectious capacity, which may also be protease sensitive, needs to be taken into account for the quantification of infectious prion formation *in vitro*. For that, unseeded and seeded (1: 100,000 dilution) substrates were submitted to a 24 h PMSA round taking samples at several time points. The soluble rec-PrP in the supernatant of the seeded sample was not detectable after 2 h (Fig 5A), indicating all PrP from the substrate had been aggregated, thus, only the samples taken at 1h and 2 h were considered for the calculation of the propagation efficiency.

Comparison of the amount of pelleted material (PrP in pellet vs. PrP in supernatant) by total protein staining (Fig 5A) shows that around 50% of the total PrP in the unseeded sample is in the pellet. This is composed entirely by non-infectious amorphous aggregates which are highly PK-sensitive (Fig 5B and 5C), providing an estimation of the maximum percentage of amorphous aggregates that could be formed in seeded reactions and which is subtracted from the total pelleted material found in seeded PMSA reactions for the final calculation of conversion efficiency. The seeded reaction reveals a faster aggregate formation, with all the PrP pelleted after 2 h of reaction. This pelleted material is purportedly composed by PK-sensitive amorphous aggregates, PK-sensitive infectious prions and infectious protease-resistant prions (Fig 5C).

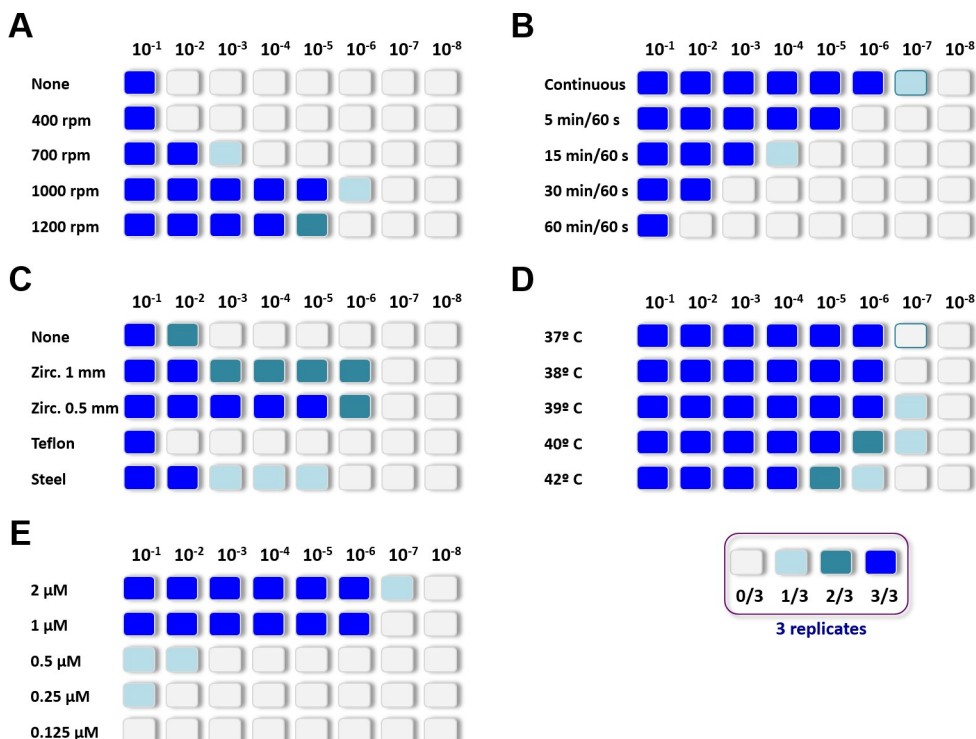

**Fig 3. Optimization of PMSA parameters to increase propagation efficiency.** Variations of several operational parameters for PMSA were tested using serial dilutions of the L-seeded-PMSA (from $10^{-1}$ to $10^{-8}$) in a substrate composed of bank vole 109I rec-PrP complemented with dextran sulfate. For each parameter variation triplicates of the dilutions were included and the maximum dilution that showed protease-resistant rec-PrP was monitored by PK digestion and Western blot after one 24 h round of PMSA. Dilutions with detectable protease-resistant rec-PrP in each triplicate are represented in different shades of blue. Studied parameters include: **(A) Shaking speed**, ranging from no shaking (none) to 1,200 rpm; optimal propagation was achieved at 1,000 rpm; **(B) Shaking/incubation intervals,** from continuous shaking to cycles of 60 s shaking and 60 min of incubation, the optimal result was with continuous shaking; **(C) Types of beads,** several beads made of different materials or sizes were tested, some previously shown to improve propagation in PMCA [84, 85]. Similar amounts of each bead type were added to the dilutions: zirconium silicate 1.0 mm and 0.5 mm, polytetrafluoroethylene (PTFE) 1.8 mm and steel 1.0 mm. Dilutions without beads were also included. Propagation in the presence of zirconium silicate beads was significantly better, beads of PTFE or steel being similar to the absence of beads. Although 0.5 mm zirconium beads were more consistent than the 1.0 mm ones both reached $10^{-6}$ maximum dilutions and thus, 1.00 mm ones were chosen due to ease of handling; **(D) Temperature** variations in the range tested from 37 to 42˚C, did not affect the propagation efficiency of seed greatly. However, propagation from the lowest dilutions was achieved at 39 and 40˚C and so 39˚C was chosen as optimum temperature setting; and **(E) PrP concentration was evaluated in substrates varying from** 0.125 µM to 2 µM. This is critical to obtain optimal amounts of material for structural studies with in vitro recombinant prion propagation methods. A balance is required between a high enough concentration to induce good propagation without wasting recombinant PrP and minimizing the amount of non-misfolded PrP remaining at the end of the reaction which will need to be eliminated prior to the structural studies. Below 1 µM PrP concentration was insufficient to efficiently propagate and the substrate with the highest protein concentration was best, so 2 µM was chosen as preferred concentration.

To calculate the percentage of PrP that misfolds into the latter two species, infectious but just partially PK-resistant *bona fide* prions, the percentage of pelleted material was estimated according to the PrP signal intensity in contrast to the amount of PrP in the supernatant and the percentage of amorphous aggregates derived from PrP in the pellet of the unseeded reaction. Therefore, considering that 100% of the PrP from the seeded reaction sediments after 2 h of PMSA and that the maximum percentage of amorphous aggregates in the unseeded reaction corresponds to 50% of the total PrP, the conversion efficiency at $10^{-5}$ seed dilution was estimated to be at least 50%. Although a greater amount of seed might reduce the formation of amorphous aggregates by driving the misfolding towards *bona fide* recombinant prion conversion pathway,

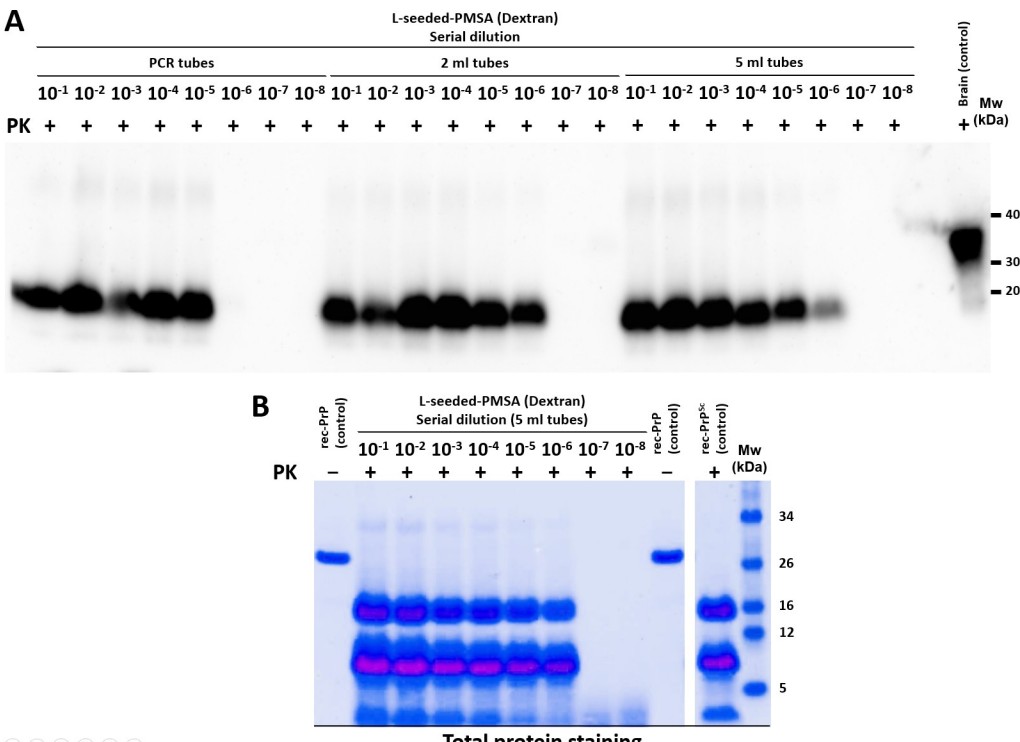

**Fig 4. Propagation of serially diluted L-seeded-PMSA using different volumes and tube sizes. A)** Western blot showing propagation capacity of L-seeded-PMSA in PCR tubes (0.2 ml) and 2 ml and 5 ml tubes using dilutions $10^{-1}$ to $10^{-8}$ after a single 24 h round of PMSA. All samples were digested at 25 μg/ml of PK. The three tube sizes allow propagation of the seed up to dilutions of $10^{-5}$–$10^{-6}$ with slightly better results in 2 ml and 5 ml tubes, demonstrating the system can be adapted to the production of large volumes. Brain (control): undigested TgVole whole brain homogenate. mAb D18 (1:5,000). **B)** Total protein staining of the serial dilution performed in the 5 ml tubes. 500 μl of each dilution were digested with PK (25 μg/ml) and concentrated by centrifugation after digestion. The resuspended pellets were subjected to electrophoresis and total protein stained with BlueSafe. Intense signal could be detected even at a dilution of $10^{-6}$ confirming the suitability of the system for propagation of large volumes. rec-PrP (control): undigested bank vole rec-PrP protein. rec-$PrP^{Sc}$ (control): original seed digested and concentrated as per the other samples.

the high dilution used for the seeded reaction allows minimization of the effect of the seed in the formation of amorphous aggregates versus *bona fide* prions. This 50% of amorphous aggregates would be the worst case scenario, in which the effect of the seed on driving the misfolding pathway towards *bona fide* prion propagation is negligible and the amorphous aggregates formed are equal to the maximum amount observed in the unseeded reaction. Nonetheless, it is likely that seeding may reduce amorphous aggregate formation and that this is an underestimation of the real conversion efficiency, which was probably greater than the 50% calculated.

Relative amounts of PK-resistant and PK-sensitive *bona fide* prions were also estimated by evaluating the self-propagation ability of the same PK-digested and undigested PMSA product ([Fig 6]). This digested or undigested seeds were serially diluted in fresh substrate and submitted to a 24 h PMSA.

The same amount of PK-treated seed showed approximately 10 to 100 folds lower propagation capacity than the undigested seed, suggesting that up to 90–95% of the seed could be composed of PK-sensitive auto-propagative prions.

## Biochemical characterization of the scaled-up seed

Prior to its biochemical characterization, the capacity for stable self-propagation of the recombinant dextran-complemented seed was evaluated. For that, 33 serial rounds of PMSA were

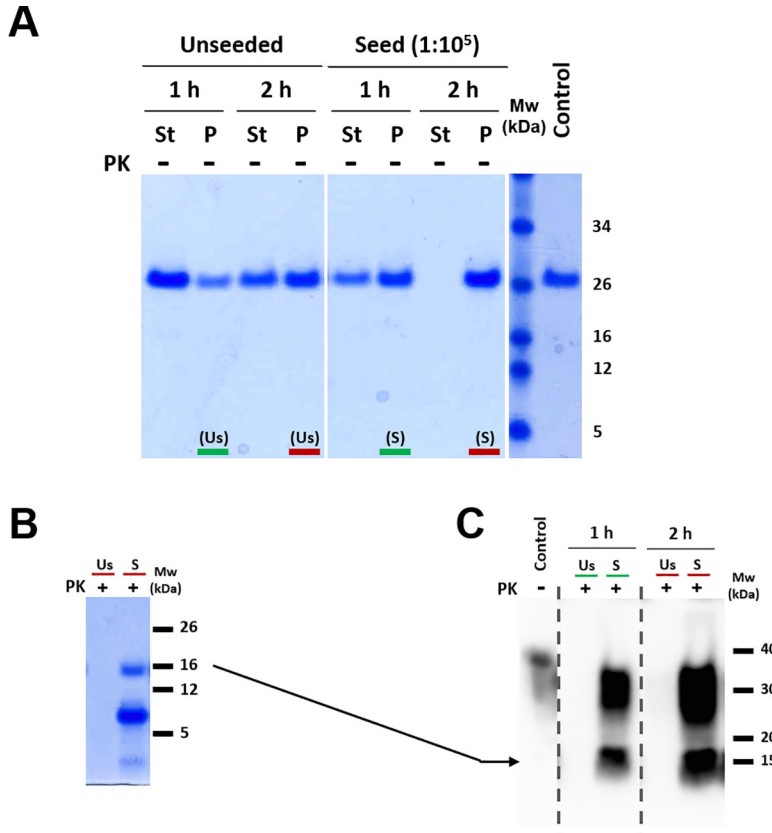

**Fig 5. Propagation efficiency in PMSA analyzed by sedimentation and protease digestion. A) Total protein staining of the undigested pellet (P) vs. supernatant (St) fractions to monitor the amounts of aggregated and soluble rec-PrP in the course of PMSA.** Fractions corresponding to 1 and 2 h from a 24 h-round of unseeded and seeded (1:100,000 dilution) PMSA reactions were collected at the indicated time points, in which all the PrP from the supernatant of the seeded sample was converted to insoluble material. Samples were centrifuged at 19,000 g for 15 min and supernatant and pellet (resuspended in a volume equal to that of the collected fraction) subjected to electrophoresis and stained for total protein with BlueSafe. The amount of aggregated PrP (pellet) and soluble PrP (supernatant) were compared in order to calculate conversion efficiency without protease digestion due to the possibility of protease-sensitive prions being an important part of the PMSA product. The unseeded reaction indicated the maximum amount of amorphous aggregates that can be formed after 2 h of the process was approximately 50% of the total PrP, which was subtracted from the aggregated fraction of the seeded reaction to estimate *bona fide* misfolding percentage. A highly diluted seed ($10^{-5}$) was used to minimize its effect on driving the misfolding process towards the *bona fide* prion formation pathway at the expense of the formation of amorphous aggregates. Therefore, it is likely that *bona fide* prion propagation efficiency is higher than the 50% estimated based on the amount of amorphous aggregates formed in an unseeded reaction. Mw: molecular weight marker. Control: total PrP on the substrate at time 0, before PMSA. **B) Total protein staining of protease digested PMSA products at 2 h.** To differentiate samples containing only amorphous aggregates (non-infectious completely protease-sensitive) from samples containing *bona fide* recombinant prions (partially protease-resistant), samples form unseeded (Us) and seeded (S) reactions were collected after 2 h of PMSA and digested with PK (25 **μg/ml) for 1h at 42˚C. As expected, the unseeded reaction contained only PK-sensitive aggregates, while the seeded one showed the characteristic PK-resistant core that remains after amino-terminal digestion, indicating that at least part of the misfolded material formed in the seeded reaction were PK-resistant *bona fide* prions. Mw: molecular weight marker. **C) Potential infectivity of unseeded and seeded samples by brain-PMCA.** In order to confirm that the unseeded sample did not contain PK-sensitive *bona fide* prions able to propagate their conformation in brain-derived PrP whereas the seeded sample was, the two samples at 2 h were used as seeds to inoculate TgVole (1x) brain homogenate at 1:10 dilution and submitted to brain-PMCA. The PMCA products were PK-digested and visualized by Western blotting, which showed the presence of PK-resistant PrP for both seeded samples while there was no detectable propagation for the unseeded ones after 1 and 2 h PMSA. The high dilution of the original L-seeded-PMSA inoculum ($10^{-5}$) protects against the possibility of being the original inoculum and not the propagated material, the result of brain PrP$^{C}$ misfolding, since the propagation capacity of L-seeded-PMSA in brain-PMCA did not reach $10^{-6}$ dilution. mAb: D18 (1:5,000). Mw: molecular weight marker. Control: undigested Tgvole (1x) PrP.

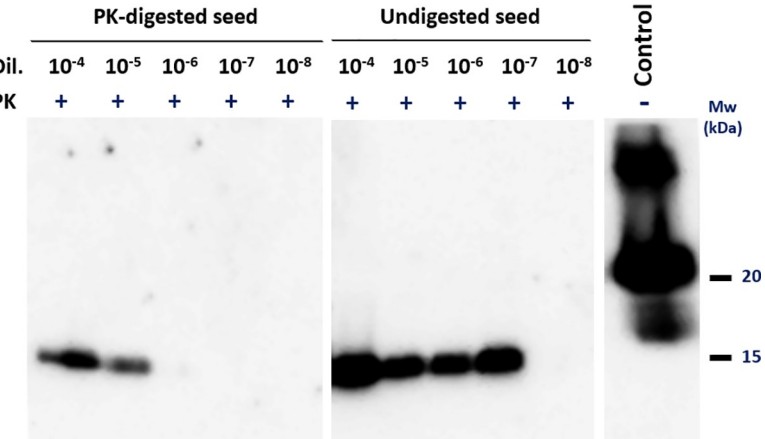

**Fig 6. Self-propagation ability of protease-digested and undigested L-seeded-PMSA.** To estimate the amount of PK-sensitive *bona fide* recombinant prion fraction in the L-seeded-PMSA sample, the propagation capacity of PK-digested and undigested seeds was compared by serial dilution (from $10^{-4}$ to $10^{-8}$) on PMSA and monitored by PK digestion and Western blot. The PK-digested L-seeded-PMSA seed showed reduced propagation capacity of approximately 1–2 logs compared to the non-digested seed, indicating that approximately 90–95% of the seed could be composed of PK-sensitive prions. mAb D18 (1:5,000). Mw: molecular weight marker. Control: undigested bank vole rec-PrP.

performed at 1:1,000 to a $10^{-100}$ dilution of the initial seed that was used for subsequent extensive biochemical characterization studies including PK digestion, electrophoresis, total protein staining and epitope mapping with several antibodies. All studies revealed no alteration of its electrophoretic migration pattern after protease digestion (Fig 7).

Total protein staining and epitope mapping of the same sample showed major bands of; ~15 kDa, corresponding to complete protease-resistant rec-PrP with partial N-terminal digestion encompassing residues ~97–231 according to the antibodies used, ~9 kDa formed by proteolytic fragments containing residues ~153–231, and ~2 kDa that includes regions 97 to ~117 of the PrP [see (→) in Fig 7C]. Another two minor bands reflected other PK cleavage sites yielding two fragments of ~12 and ~6 kDa [see green and red bands in Fig 7C]. To evaluate the PK resistance, equal amounts of the misfolded recombinant PrP were digested with increasing concentrations of proteinase-K (from 50 to 1,000 µg/ml; rec-PrP:PK ratio from 1:2 to 1:40) at 42°C for 1 h followed by visualization by electrophoresis and total protein staining (Fig 7B). At 1,000 µg/ml of PK digestion was incomplete indicating high resistance to protease characteristic of many prion isolates.

Analysis of PK-treated PMSA product by ESI-TOF allowed the exact identification of several PK-resistant fragments (Supplementary information, S1 Table) and confirmed that PK cleaves between positions $H_{96}/N_{97}$ and $N_{97}/Q_{98}$ generating amino-terminal ragged ends, and also vigorously between positions $E_{152}/N_{153}$ and $N_{153}/M_{154}$ generating PK-resistant N- and C-terminal shorter fragments of ~5–6 and ~9 kDa, respectively. Very minor peaks corresponding to additional minor cleavages at $A_{116}/A_{117}$, $S_{132}/A_{133}$ and $M_{134}/S_{135}$ were also detected. These data are consistent with results obtained from SDS-PAGE/total protein staining and epitope mapping analyses (Fig 7C). The higher sensitivity of WB and the large amount of loaded sample allowed identification of some additional PK cleavage sites within the misfolded PrP propagated by PMSA that were not observed previously in the WB done with the usual sample amounts.

In order to gain some insight on the ultrastructural characteristics of the L-seeded-PMSA sample, the size distribution of the aggregates was evaluated by zonal sedimentation analysis

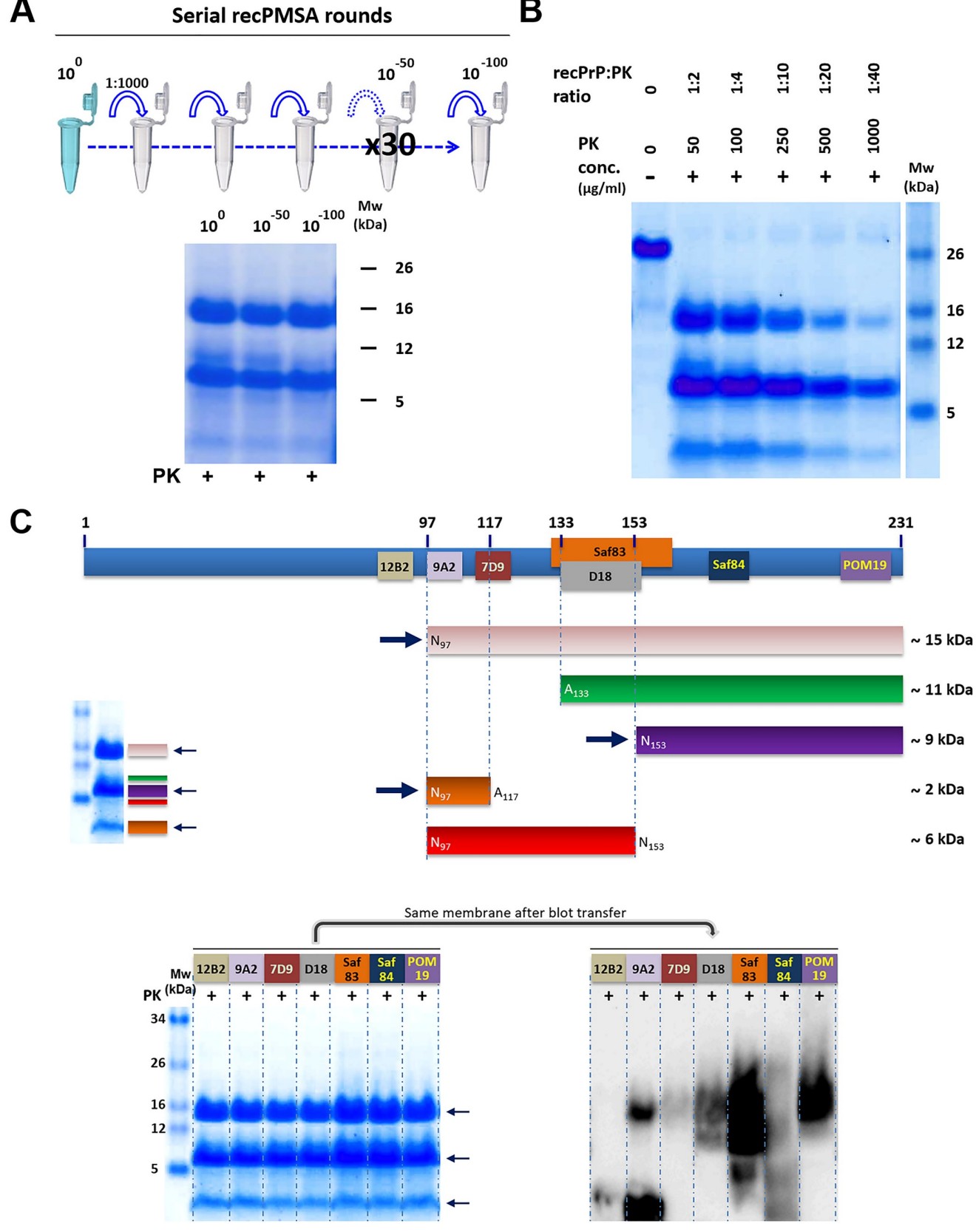

**Fig 7. Biochemical characterization of L-seeded-PMSA. A)** The ability of L-seeded-PMSA to self-propagate indefinitely shown through 30 serial rounds of PMSA at 1:1,000 dilution. 500 µl of the original seed ($10^0$ dilution), the product of round 17 of PMSA ($10^{-50}$ dilution) and the product of round 33 ($10^{-100}$ dilution) were PK-digested, concentrated by centrifugation and visualized by total protein staining and showed no detectable pattern differences throughout the propagation process. **B)** Protease-K (PK) resistance of L-seeded PMSA was evaluated by submitting fractions of the same sample to increasing PK concentrations for 1h at 42°C. L-seeded-PMSA showed high resistance to protease, as the main PK-resistant bands are maintained even at the highest PK concentration used. **C)** Identification of the proteolytic fragments of L-seeded-PMSA by epitope mapping was performed with the same PK-digested and concentrated sample divided in seven that were stained for total protein to guarantee equal protein amounts in each lane. The stained gel was then transferred for Western blotting and the following antibodies were used for epitope mapping: 12B2 (1:2,500), 9A2 (1:4,000), 7D9 (1:1,000), D18 (1:5,000), Saf83 (1:400), Sa84 (1:400) and POM19 (1:10,000). The theoretical reconstruction of the PrP fragments recognized by each antibody are represented in the cartoon, the most abundant fragments were (see arrows) ~15 (pink), ~9 (purple) and ~2 (orange) kDa. Another two minor bands reflect other PK cleavages yielding two fragments of ~12 and ~6 kDa (green and red bands, respectively). These fragments are not always detected by all the antibodies used for the epitope mapping due to their low amounts, below the detection limit of some of the antibodies. MW: Molecular weight.

on continuous sucrose density gradient. The sample produced in PMSA, either undigested or digested with PK, was submitted to ultracentrifugation through 10–80% sucrose gradient to determine the degree of quaternary structure heterogeneity by total PrP or PrP$^{res}$ detection in Western blot (Supplementary information, S3 Fig). A continuous distribution of PrP aggregates is observed starting in fractions with around 30–35% sucrose to the end of the gradient with 80% sucrose with no monomeric PrP in upper fractions. This suggest that all PrP is misfolded during the process forming mostly large fibrillary aggregates with quite heterogeneous sizes, densities ranging from 1.1 g/cm$^3$ to higher than 1.3 g/cm$^3$ [44]. The analysis of PK-treated sample reveals a similar sedimentation velocity pattern, with a continuous distribution of aggregates from 40–45% sucrose to 80%, indicating that part of the smallest aggregates could be PK-sensitive or composed by amorphous aggregates. In any case, the linearity of the signal and the major signal intensity towards the bottom fractions clearly indicates that the sample is mainly composed by very big PK-resistant aggregates able to reach and even cross the 80% sucrose gradient, although smaller PK-resistant aggregates are also detected as proof of some quaternary structure heterogeneity dominated by large aggregates.

## Biological characterization of the scaled-up seed

**Infectivity and transmissibility in animal models.** The definitive demonstration of the infectivity of the recombinant seed propagated by PMSA, in comparison with the original inoculum generated by PMCA, was done by intracerebral inoculation in TgVole (1x) mouse and bank vole models, both bearing bank vole I109 PrP with similar expression levels (Supplementary information, S2 Fig).

In TgVole (1x) model, which is very similar to bank voles as shown by similar short incubation periods for CWD-vole prion strain [58±1 days post inoculation (dpi) in TgVole and 35–40 dpi in bank voles] in both models [18], the original PMCA-derived seeds caused clinical disease in all inoculated animals with incubation periods of 175±6 (*Prnp$^{0/0}$* complemented) and 326±9 dpi (dextran sulfate complemented). Two different preparations of the PMSA-adapted seed showed the same attack rate (100%) with incubation times of 250±4 and 266±3 dpi. Negative control groups inoculated with non-misfolded rec-PrP and with the same rec-PrP fibrillated following another protocol [45] showed no signs of disease after > 500 dpi, as expected (Table 1).

A second passage was performed using TgVole (1x) animals, which showed shorter incubation periods of 120±1 dpi for the PMCA-derived seed and of 116±2 dpi for the PMSA-derived one. A biochemical analysis of Proteinase-K (PK)-resistant PrP$^{Sc}$ in brain homogenates from these inoculated TgVole (1x) showed that the CWD-vole and all the *bona fide* misfolded rec-PrPs accumulated in the classical PrP$^{Sc}$ three-banded electrophoretic migration pattern. However, brains from animals inoculated with the non-infectious fibrillary rec-PrP and non-fibrillated rec-PrP were devoid of any PK-resistant bands (Fig 8).

**Table 1. First passage of PMCA and PMSA-adapted recombinant samples inoculated into TgVole (1x) mice.**

| Inoculum | Substrate | Model | Survival time of positive animals (dpi) (mean±SEM) | Attack rate[a] |
|---|---|---|---|---|
| CWD-vole | Brain | TgVole | 58±1 | 10/10 (100%) |
| H/L-seeded-03 | Brain | TgVole | 175±6 | 8/8 (100%) |
| L-seeded-PMCA | PMCA | TgVole | 326±9 | 5/5 (100%) |
| L-seeded-PMSA (1)[b] | PMSA | TgVole | 266±3 | 9/9 (100%) |
| L-seeded-PMSA (2)[b] | PMSA | TgVole | 250±4 | 10/10 (100%) |
| Fibrillary vole rec-PrP | Urea | TgVole | >500 | 0/6 (0%) |
| Non-fibrillary vole rec-PrP | - | TgVole | >500 | 0/6 (0%) |

[a] Data obtained based on clinical signs and protease-resistant PrP detection.

[b] (1) and (2) indicate two different preparations of L-seeded-PMSA done independently by PMSA propagation of L-seeded-PMCA in a substrate based on recombinant bank vole I109 PrP.

SEM: Standard Error of Mean. dpi: day post inoculation.

Regarding bank vole I109, the same PMCA and PMSA-derived seeds caused disease in 239 ±46 dpi [10] and 318±3 dpi, respectively, again with 100% attack rates (Table 2).

The biochemical and immunohistopathological examination of the brain samples from inoculated bank voles shows classical PK-resistant PrP and clear PrP^Sc plaque deposition (Fig 9), confirming the *bona fide* prion disease induced by L-seeded-PMSA inoculum.

Both seeds were able to cause disease in all the inoculated animals, either bank voles or transgenic mouse models with no PrP overexpression, with similar incubation times and complete attack rate. Therefore, the PMSA propagates this recombinant infectious prion faithfully, providing material suitable for structural studies.

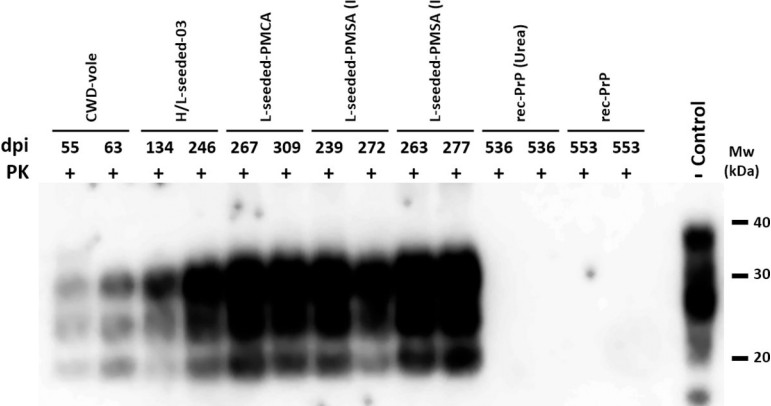

**Fig 8. PrP^Sc detection in diseased TgVole (1x) brains inoculated with CWD-vole, H/L-seeded-03, L-seeded-PMCA, L-seeded-PMSA, non-infectious fibrillary rec-PrP and non-fibrillated rec-PrP.** Biochemical analysis of Proteinase-K (PK)-resistant PrP^Sc in brain homogenates from TgVole (1x) inoculated with different misfolded PrPs: CWD-vole, H/L-seeded-03, L-seeded-PMCA, L-seeded-PMSA (I & II), non-infectious fibrillary rec-PrP and non-fibrillated rec-PrP. Representative TgVole brain homogenates were digested with 200 μg/ml of PK. The CWD-vole and the PMCA and PMSA-derived misfolded rec-PrPs inoculated TgVole brains accumulated a classical PrP^Sc type characterized by a three bands electrophoretic migration pattern. Brains from animals inoculated with the non-infectious fibrillary rec-PrP and non-fibrillated rec-PrP, used as negative controls, did not show any PK resistant band. D18 monoclonal antibody (1:5,000). Control: undigested TgVole (1x) whole brain homogenate. MW: Molecular weight. dpi: days post-inoculation at which each animal was culled due to neurological clinical signs.

**Table 2. First passage of PMCA and PMSA-adapted recombinant samples inoculated into bank voles.**

| Inoculum | Substrate | Model | Survival time of positive animals (dpi) (mean±SEM) | Attack rate[a] |
|---|---|---|---|---|
| CWD[b] | Brain | Bank vole | 37±1 | 9/9 (100%) |
| L-seeded-PMCA[c] | PMCA | Bank vole | 239±46 | 8/8 (100%) |
| L-seeded-PMSA (1) | PMSA | Bank vole | 318±3 | 4/4[d] (100%) |

[a] Data obtained based on clinical signs and protease-resistant PrP detection.

[b] Data obtained and published previously using CWD: Chronic Wasting Disease adapted to bank vole I109 (10).

[c] Data obtained and published previously using the dextran-complemented recombinant sample (10).

[d] From a group of eight animals, four had to be killed due to unrelated intercurrent disease and were excluded from these calculations.

SEM: Standard Error of Mean. dpi: day post inoculation.

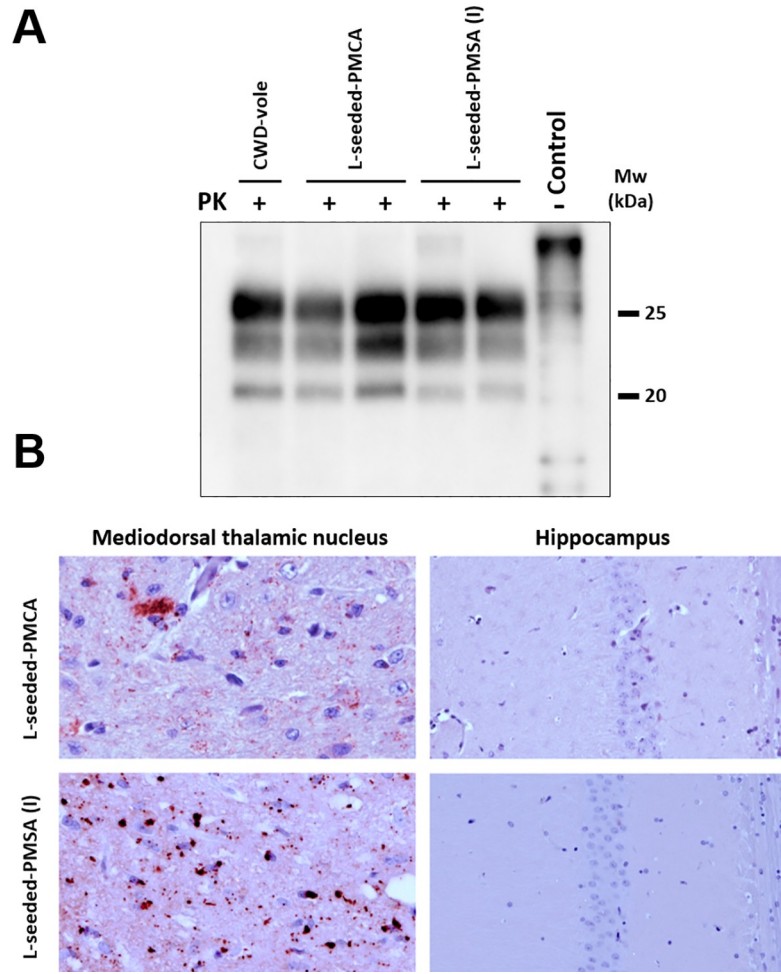

**Fig 9. PrP^Sc detection and histopathological analysis of diseased bank vole 109I brains inoculated with CWD-vole, L-seeded-PMCA and L-seeded-PMSA. A)** Biochemical analysis of Proteinase-K (PK)-resistant PrP$^{Sc}$ in brain homogenates from bank vole I109 inoculated with the misfolded rec-PrPs: L-seeded-PMCA and L-seeded-PMSA, and CWD-vole as control. Representative bank vole brain homogenates were digested with PK (200 μg/ml). All rec-PrP$^{Sc}$ inoculated bank vole brains accumulated a classical PrP$^{Sc}$ type characterized by a three-banded electrophoretic migration pattern, indistinguishable from the CWD-vole inoculated vole brains. 9A2 monoclonal antibody (1:5,000). Control: undigested bank vole whole brain homogenate. MW: Molecular weight. **B)** Brain deposition of PrP$^{Sc}$ in the mediodorsal thalamic nucleus and the hippocampus of bank voles inoculated with L-seeded-PMCA or L-seeded-PMSA was assessed by immunohistochemistry using the monoclonal antibody 6C2 (1:300). Note, with both seeds, deposition of PrP$^{Sc}$ could be observed in the mediodorsal thalamic nucleus but not in the hippocampus.

**Specific infectivity of L-seeded-PMSA.** The end-point *in vivo* titration and determination of specific infectivity was assessed in TgVole using tenfold dilutions of L-seeded-PMSA being $10^{-4}$ (Table 3).

From these data, the specific infectivity of L-seeded-PMSA in TgVole (1x) was also calculated using the Spearman-Karber method [46]. Taking into account that the dose causing fatal disease in 50% of the animals ($LD_{50}$) is between dilutions $10^{-4}$ and $10^{-5}$, of which 10 μl were inoculated with a total PrP content of 0.25 μg, specific infectivity was calculated to be of 6.34 x $10^4$ / μg of PrP in TgVole (1x).

## Preliminary ssNMR study of the recombinant prion

With this highly scalable method that is able to propagate a *bona fide* recombinant prion, preparing up to 400 ml of misfolded seed per day is easily feasible in any laboratory requiring large amounts of material for structural studies. As proof of concept, a preliminary ssNMR experiment was conducted using 400 ml of a PMSA product containing dual isotopically labeled bank vole I109 rec-PrP ($C^{13}$ and $N^{15}$), seeded with unlabeled L-seeded-PMSA 1:100 and prepared as follows. To obtain a misfolded sample to act as a seed, with at least 99.99% of the protein labeled, two serial rounds of PMSA were performed using in the first passage, the non-labeled L-seeded-PMSA seed diluted 1:100. Subsequently, 4 ml of the isotopically labeled seed was scaled 100-fold up to reach, in a single round (24 h), 400 ml of misfolded material labeled at 99.9999%. With an estimated efficiency of conversion of at least 50%, approximately 9 mg of converted material were obtained from the 400 ml at 2 μM rec-PrP concentration (containing around 18.4 mg of rec-PrP). However, the final amount of PK-digested material was approximately 0.75–1.0 mg, indicating that around 90–95% of the misfolded material was PK-sensitive, in agreement with the ability of the PK-undigested L-seeded-PMSA seed to propagate by PMSA 10–100 times (1–2 logs) more than when is PK-digested. In order to reduce the 400 ml of seed produced by PMSA to the standard ssNMR working volumes of 10–100 μl, the sample was digested with PK (25 μg/ml for 1 h at 42°C), based on the previous characterization. After digestion, the sample was concentrated by centrifugation.

An INEPT spectrum recorded at 15°C nominal temperature did not show any signals, demonstrating that misfolded rec-PrP^Sc does not contain any highly flexible parts [47]. Conversely, $^{13}$C CP/MAS based NMR spectra which exhibit signals from immobilized parts of the protein, showed the full set of resonances. Additionally, two-dimensional NMR-spectroscopy was possible (Fig 10) in which signal sets for the spin systems of the amino acid types alanine, threonine, isoleucine and valine were clearly identified.

The preliminary results obtained from the ssNMR spectra demonstrate that large amounts of isotopically labeled prions suitable for this and other biophysical methods can be propagated by PMSA, to enable elucidation of the three-dimensional structure of prions or at least of the most relevant structural motifs.

**Table 3. Titration of the PMSA-adapted recombinant samples inoculated into TgVole (1x).**

| Inoculum | Dilution | Model | Survival time of positive animals (dpi) (mean±SEM) | Attack rate[a] |
|---|---|---|---|---|
| L-seeded-PMSA (1) | $10^{-1}$ | TgVole | 266±3 | 10/10 (100%) |
| L-seeded-PMSA (1) | $10^{-2}$ | TgVole | 281±6 | 4/4 (100%) |
| L-seeded-PMSA (1) | $10^{-4}$ | TgVole | 321±6 | 5/5 (100%) |
| L-seeded-PMSA (1) | $10^{-5}$ | TgVole | 434±21 | 2/5 (40%) |

[a] Data obtained based on neurological clinical signs and protease-resistant PrP detection.

SEM: Standard Error of Mean. dpi: day post inoculation.

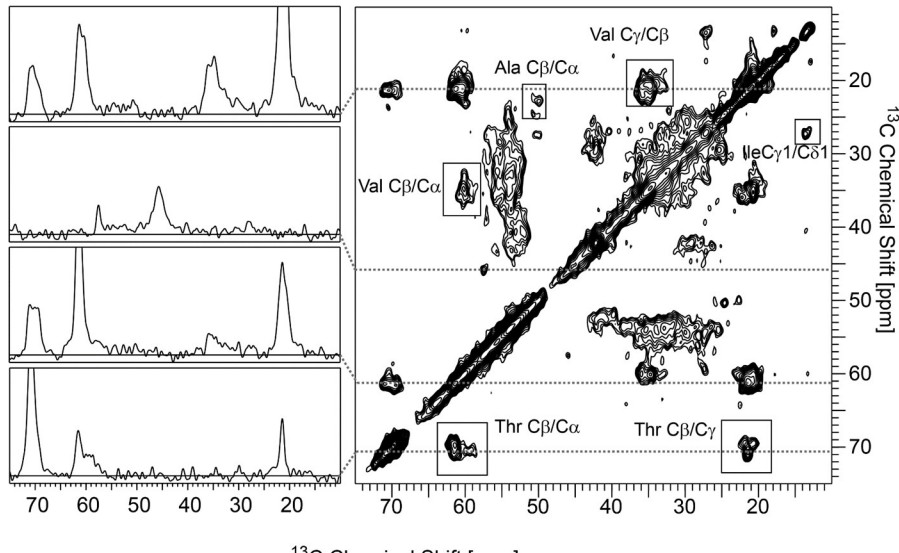

**Fig 10. ssNMR.** Right side: 2D $^{13}$C/$^{13}$C correlation spectrum employing proton driven spin diffusion (50 ms) for homonuclear polarization transfer. Cross correlation signals of selected amino acids are highlighted in the spectrum. On the left side, four representative 1D slices through the spectrum are displayed. The spectrum was recorded at a nominal temperature of 0°C and a field strength of 14.1 T with a spinning frequency of 11 kHz.

## Discussion

Deciphering the three dimensional structure of prions at high resolution has been long hindered by several characteristics of these proteins. Isolation and purification of PrP$^{Sc}$ from brain homogenates in sufficient quantities required for most of the high-resolution biophysical techniques is extremely difficult, despite advances in GPI-deficient PrP-expressing animal models [34]. Although *in vitro* prion conversion assays and the use of recombinant PrP addressed many of the purification issues [1–10], that PrP$^{C}$ and, in particular, rec-PrP can misfold into multiple different conformations and aggregated states (amorphous, amyloid fibrils or 2D crystals) with some unrelated to disease or some disease-associated but not transmissible, complicates structural studies [48]. Even among transmissible, diseases-associated aggregates, known as PrP$^{Sc}$, the co-existence of distinct strains and the variety of quaternary structures ranging from oligomers to fibrils with different sizes make structural characterization of prions extremely challenging [37]. Considering the variety of misfolded forms PrP can adopt and the lack of knowledge on the event itself, it is not unexpected that the propagation of recombinant prions *in vitro* resulted in many different products ranging from biologically inert aggregates to highly transmissible and neurotoxic species or *bona fide* prions [1–10, 12, 15, 21]. We and others have previously reported the generation of recombinant prions using PMCA and *Prnp$^{0/0}$* brain homogenates or specific cofactors [5, 8–10, 14, 15, 49]. Our group also proposed a way to reduce the heterogeneity of these *in vitro* generated prions by propagation in the presence of specific cofactors [10]. However, lack of knowledge of the molecular events underlying prion propagation by PMCA and the difficulty controlling sonication parameters render highly variable results; i. e. non-infectious misfolded rec-PrP was produced with the exact conditions that had previously yielded a highly infectious recombinant prion [12, 50]. Additionally, the complexity of the equipment necessary for PMCA inherently limits scalability and generation of large amounts of misfolded material. Herein we present a new system that overcomes all the issues of complexity, variability and scalability of PMCA, which allows generation of suitable material for high-resolution protein structural studies.

Fundamentally based on the PMCA: with the same substrate preparations complemented with a single cofactor (dextran sulfate) for recombinant prion propagation [10], zirconia/silica beads shown previously to enhance the performance of classical brain-homogenate based PMCA [51] and reaction temperature, the critical difference is the substitution of sonication by shaking, hence the name Protein Misfolding Shaking Amplification (PMSA). Although shaking has been used extensively in several cell-free systems for amyloid formation or prion propagation, none showed the capacity to yield recombinant prions capable of causing classical clinical prion disorder with a 100% attack rate, neurological signs, spongiform change and misfolded PrP accumulation in the CNS upon first passage in wild-type animal models and transgenic models devoid of *PRNP* gene overexpression [1, 5, 52]. Contrary to other methodologies and previous assumptions, PMSA shows that, under appropriately optimized conditions, vigorous shaking provides sufficient energy to achieve *bona fide* prion propagation in the presence of an adequate concentration of the polyanionic cofactor dextran sulfate. Thus, the idea that high energy input, as purportedly provided by sonication in PMCA, could be required to cause fibril breakage and efficiently misfold PrP into its infectious counterpart *in vitro* needs to be reexamined [53].

The adaptation of an infectious recombinant prion generated by PMCA to this new PMSA system was successful and seemed to conserve biochemical features, as indicated by their electrophoretic migration patterns after PK digestion and the ability to propagate on brain-based substrates in a single round of brain-PMCA. In particular, the PK fragmentation pattern of L-seeded-PMSA was consistent with those reported for its precursor obtained by PMCA [33]. MALDI analysis of such samples produced spectra that were dominated by a large peak corresponding to fragment $N_{153}$-$S_{231}$ and a group of peaks that were believed to correspond to doubly N- and C-terminal-truncated fragments which the lower mass accuracy of MALDI did not allow to identify, some of which would correspond to $N_{97}$/$Q_{98}$-$E_{152}$/$N_{153}$. Very small peaks, corresponding to additional minor cleavages at $A_{116}$/$A_{117}$, $S_{132}$/$A_{133}$ and $M_{134}$/$S_{135}$, were detected also. Conversely, some minor differences might exist between the two products. This pattern of fragmentation, as discussed in our previous publication [33] strongly suggests that both PMCA- and PMSA-generated recombinant PrP$^{Sc}$ share the same architecture. Besides the main cleavage at $N_{97}$/$Q_{98}$, which marks the division between the disordered N-terminal tail and the compact 'PrP27-30' domain, a significant secondary cleavage site was seen for both brain-derived and recombinant PrP$^{Sc}$ at $N_{153}$-$M_{154}$ [32]. In recombinant prion samples, cleavage at this site was much more extensive than in brain-derived PrP$^{Sc}$, and the resultant N- and C-terminal-truncated fragments were detected readily. We have proposed previously that the secondary cleavage site, which is close to proline $P_{158}$ might correspond to a particularly accessible flexible linker connecting β-strands [54]. Other shared minor cleavage sites also support the notion that brain-derived PrP$^{Sc}$ and the recombinant prion described here are comprised of a set of β-strands connected by flexible linkers, spanning from position 90–98 to the C-terminus, and that the identity of the β-strands and linkers are similar in both molecules. The data from epitope mapping and ESI-TOF of the proteolytic fragments of this recombinant prion could fit with one of the suggested structural models for prions, the 4-rung β-solenoid model [34, 48], according to the scheme proposed (Fig 11). However, it does not unequivocally exclude the possibility of other structures giving rise to the same proteolytic fragments.

Another possibility that we cannot completely exclude, although it contradicts Occam´s principle, is that a mixture of strains is responsible for the multiple-banded pattern, some proteolytic fragments originating from certain conformer, while other fragments would derive from different structural variants. Our attempts to separate this possible strain mixture through clonal selection of limiting dilutions (highest dilution able to propagate in each PMSA round was diluted from $10^{-1}$ to $10^{-8}$ in a subsequent round and this process was repeated

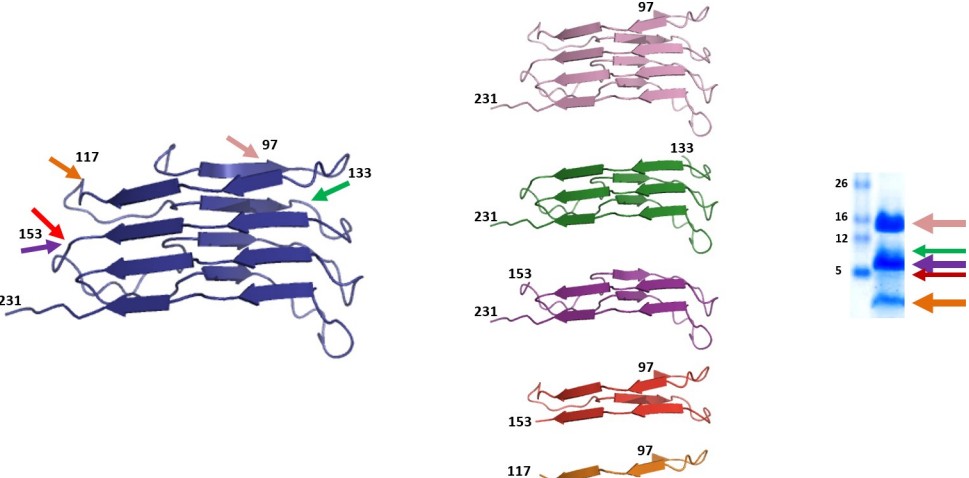

**Fig 11. Graphical representation of the PrP fragments identified after PK-digestion of L-seeded-PMSA by epitope mapping and ESI-TOF and their origin based on the 4 rung β-solenoid model for PrP<sup>Sc</sup>.** The identification of the main proteolytic fragments of L-seeded-PMSA after PK digestion (characteristic whole PK-resistant core after amino terminal digestion represented in blue, encompassing residues 89–231) by epitope mapping and ESI-TOF showed four major cleavage sites at positions ~97, ~117, ~133 and ~153 that give rise to five fragments of ~15 (pink, encompassing residues ~97–231, rungs 1 to 4), ~11 (green, encompassing residues ~133–231, rungs 2/3 to 4), ~9 (purple, encompassing residues ~153–231, rungs 3 and 4), ~6 (red, encompassing residues ~97–153, rungs 1 to 3) and ~2 (orange, encompassing residues ~97–117 rungs 1 and 2) kDa. Based on the assumption that PK cleaves accessible and flexible sites of the protein, most likely loops linking β-strands, the main fragments and cleavage sites may indicate that the overall structure of L-seeded-PMSA could fit with the 4-rung β-solenoid model proposed previously for brain-derived prions [34].

along more than 5 serial rounds) always resulted in the same proteolytic fragments, suggesting a single conformer or the existence of multiple conformers with exactly the same propagation abilities and thus, undistinguishable by this method.

The sedimentation velocity analysis could also shed some light on the heterogeneity of the sample produced by PMSA, mainly in terms of quaternary structure heterogeneity. This methodology has been applied previously to establish the relation between infectivity and specific prion aggregates [55], to determine the heterogeneity of PrP assemblies in prions from brains [56] and to characterize prion strains based on the differential properties of the assemblies present in each isolate [57]. The ultrastructural heterogeneity on samples containing PrP assemblies, either recombinant or brain-derived, can indicate the presence of a continuous range of aggregates with the same tertiary structure but different fibril sizes or the existence of structurally distinct conformers that adopt slightly different quaternary structures. The continuous distribution for aggregates observed with L-seeded-PMSA, with predominance of very large aggregates and similar pattern of PK-digested and undigested samples (Supplementary information, S3 Fig), together with the other biochemical characteristics evaluated, suggests a single population with a unique tertiary structure forming assemblies of distinct sizes. Nonetheless, the existence of slightly different tertiary structures giving rise to distinct assemblies cannot be completely discarded with the present evidences. The first scenario should not pose a great limitation for the ssNMR studies aimed to elucidate the tertiary structure of our recombinant PrP<sup>res</sup> preparations, since the main problems associated to sample heterogeneity are not related to the size of the assemblies (i. e. number of monomers within a fibril). As long as the tertiary structure is homogeneous the large majority of monomers within the fibrils will have the same surrounding (same distances between residues) and the few molecules at the fibril ends will be too few to be detected. So the average fibril length is not necessarily

correlated with the monomer structure and thus, variations in particle size *per se* (considering the same structure for monomers forming them) would not affect the quality of the NMR spectra. The proof of the relative importance of fibril-length heterogeneity is provided by other amyloid structures solved using preparations with variable length fibrils and oligomers [58]. In the less probable second scenario, the different monomeric structures within the sample would be critical for structural resolution by ssNMR but this would also lead to structural polymorphism (i.e. different fibril morphologies like straight/twisted and it may also be accompanied by different numbers of monomers per layer). Presumably, such morphological differences would be reflected on different majoritarian populations with distinct density in the sedimentation velocity profile. The issue could also be addressed by negative stain EM [59] but only if differences were quite notorious. Moreover, despite fibril polymorphism can be associated with different monomer folds, fibrils with very different fiber morphology but still the identical monomer conformations have also been demonstrated [60]. In any case, any effect of the morphological variability of the PrP fibrils will be continuously monitored during the ssNMR spectra generation, in order to introduce further improvements in sample preparations, if required, to solve issues related to quaternary structure of assemblies.

Regarding the biological properties of L-seeded-PMSA upon inoculation in TgVole (Table 1) and bank voles (Table 2), there is no doubt about its infectivity and the capacity to cause a *bona fide* prion disease. It is noteworthy that differences in incubation times were detected between TgVole and bank voles in the first passage of L-seeded-PMSA, despite the identical PrP sequence and very similar expression levels (S2 Fig). Although unexpected, this difference might be explained by the differences between both models and the existence of a transmission barrier between the recombinant inoculum and the PrP$^C$ from brain. The significant reduction of disease incubation times in a second passage observed in TgVole for L-seeded-PMSA (from 266–250 dpi to 116 dpi) and also for L-seeded-PMCA (from 326 dpi to 120 dpi) strongly suggest the existence of a transmission barrier, most likely due to the absence of glycosylation and GPI anchoring of the recombinant inoculum. Usually, a transmission barrier is associated to prolonged incubation times upon first passage and higher dispersion in disease onset and duration between animals of the same group [61], but may also enhance the effect of small differences in PrP expression levels and the genetic background of the models used to evaluate infectivity. This could be the reason for the differences observed in incubation times, since small differences in expression levels undetectable by western blotting are plausible and genetic background differences obvious being two different species. In any case, L-seeded-PMSA with 100% attack rates in both models, is undoubtedly infectious and gives rise to a classical prion disorder with all its hallmarks.

Despite the possibility of strain mixtures could not be definitively excluded, the stable serial propagation of this prion over 33 rounds of PMSA at a dilution of 1:1,000 demonstrated its stability and therefore suitability for high resolution structural studies, demonstrated further by a good ssNMR spectra being obtained. Although the signal dispersion was limited in the preliminary spectra shown in this manuscript (Fig 10), such that site-specific resonance assignment was not feasible, it can clearly be seen that most of the cross-correlation signals have chemical shifts which are indicative of β-strand-like conformations, in agreement with the β-sheet rich secondary structure of amyloid fibrils. However, single residues may have random-coil or even α-helical conformation. Further structural characterization, such as the determination of the registry and supramolecular arrangement of the β-strands within the fibril will be possible despite a lack of a site-specific resonance assignment and in the presence of some degree of structural heterogeneity if dedicated labeling schemes and pulse sequences for distance measurements are applied [62–64].

The titer of recombinant prions produced *in vitro* has always been a concern regarding the biological significance of the samples analyzed and their resemblance with brain-derived

prions. The possibility of obtaining large amounts of non-infectious self-propagative aggregates that could resist PK treatment together with *bona fide* prions or the existence of heterogeneous strain mixtures are complicated issues to assess. The specific infectivity of our intracerebrally inoculated L-seeded-PMSA preparation according to the Spearman-Karber method is $6.34 \times 10^4$ LD$_{50}$ / µg of PrP in TgVole (1x). This is similar, in fact slightly higher, than that obtained for the first highly infectious synthetic prion generated with RNA and POPG as cofactors in wild type mice ($10^4$ LD$_{50}$ / µg of PrP) [65]. More recently, the same authors reported on a recombinant mouse PrP$^{Sc}$ also made with RNA and POPG as cofactors, with higher specific infectivity, namely, $2 \times 10^5$ LD$_{50}$ / µg of PrP; however this was assessed in a transgenic mouse model (Tga20) that overexpresses mouse PrP at 8x the level of a wild type mouse and has repeatedly shown to exhibit increased susceptibility to prion infection compared to wild type animals [66], therefore, comparison is not straightforward. Other phospholipid-complemented preparations have been reported with specific infectivity of $2.2 \times 10^6$ LD$_{50}$ / ug of PrP [15], in this case tested in wild type models, and $5.6 \times 10^5$ LD$_{50}$ / µg of PrP using the same substrate preparation but a different *in vitro* propagation method [67]. All in all, the specific infectivity of our preparation lies within the range of values reported. It should be noted that while differences in infectivity titer of different samples of a given PrP$^{Sc}$ strain are a direct indication of purity and structural integrity, comparing specific infectivity of different PrP$^{Sc}$ strains is problematic, as different strains exhibit not just different incubation times but also different specific infectivity, a classic example being the different properties of Drowsy and Hyper [68]. Furthermore, the propagation dynamics can be decisively altered by the quaternary structure that each strain or even specific preparation of a given strain acquires *in vitro* in comparison with the behavior of PrP$^{Sc}$ formed or transmitted in the encephalon [69]. It should be noted also, that specific activity is usually lower in recombinant prions due to lack of post-translational modifications [49, 70]. All of the above likely explains the range of specific infectivity reported for different strains of infectious recombinant PrP$^{Sc}$ which, on the other hand, differ clearly from the several reported propagative but non-infectious PrP samples, whose specific infectivity is essentially zero [12, 21]. In this respect, it is noteworthy that in contrast to other methods reported [12, 21, 45, 49], we have never observed generation of similar non-infectious propagative PrP species with our method. In fact, we have purposefully tried by introducing an array of different minor changes in the experimental conditions, with the objective of obtaining such material for comparative purposes, and have never been able to propagate such non-infectious conformers. This, together with the sedimentation velocity analysis and our cloning studies (*vide supra*) makes very unlikely the possibility that our material consists of a mixture of a small quantity of a highly infectious material and a bulk of "inert", non–infectious material, a very important consideration for structural studies.

Overall, we demonstrate here that shaking can be used to continuously propagate infectious recombinant prions as long as appropriate conditions and specific cofactors, such as dextran sulfate, are used. Moreover, we present a set of tools to extensively characterize misfolded PrP species generated *in vitro*, including a method to evaluate potential infectivity *in vitro* (brain-PMCA) prior to the more costly and protracted bioassays. The simplicity, scalability and robustness of this novel propagation method makes it easily implementable in any laboratory interested on generating large amounts of recombinant prion samples which is likely to be successful for other recombinant prions previously reported to propagate *in vitro* in presence of single cofactors [8, 10, 14, 15, 21]. This system might be adapted to the propagation of other prion strains apart from the one presented here and may open the way to detailed structural studies on many different prions shedding light on the strain phenomenon. Importantly, PMSA can become an invaluable source of distinct recombinant prions that can be labeled in multiple ways and provide the amounts (in the milligram range) of infectious prions required

for one of the most critical open questions underlying prion biology, the atomic structure of this extraordinary pathogen.

## Materials and methods

### Preparation of purified recombinant PrP

Bacterial expression and purification of bank vole I109 recombinant PrP (amino acids 23–231) (rec-PrP) was performed as described previously [71]. Briefly, pOPIN E expression vector containing the wild type I109 bank vole *Prnp* gene was prepared by standard molecular biology techniques using the oligonucleotides 5' AGGAGATATACCATGAAGAAGCGGCCAAAG CCTGG3' and 5' GTGATGGTGATGTTTGGAACTTCTCCCTTCGTAGTA3' to derive the PrP of interest from genomic DNA of bank vole I109 and clone in the pOPIN E expression vector. *E. coli* Rosetta (DE3) Competent Cells (EMD Millipore) were transformed with the expression vector using standard molecular biology procedures allowing the expression of the recombinant protein in LB broth (Pronadisa) upon Isopropyl β-D-1-thiogalactopyranoside (IPTG) (Gold biotechnology) induction. Although the protein does not contain a His-tag, purification of the protein was performed with a histidine affinity column (HisTrap FF crude 5 ml, GE Healthcare Amersham) taking advantage of the natural His present in the octapeptide repeat region of PrP. After elution in buffer (20 mM Tris-HCl, 500 mM NaCl, 500 mM imidazole and 2 M guanidine-HCl, pH 8), the quality and purity of protein batches was assessed by BlueSafe (NZYtech) staining after electrophoresis in SDS-PAGE gels (BioRad). Finally, guanidine-HCl was added, to a final concentration of 6 M, for long-term storage of purified protein preparations at -80˚C. For uniformly $^{13}$C, $^{15}$N labeled recombinant bank vole I109 PrP preparation, the same vector and *E. coli* strain were used growth in minimal medium composed of M9 minimal medium (Thermofisher Scientific), $^{13}$C-glucose (3 g/liter) (Eurisotop, Cambridge Isotope Laboratories) and $^{15}$NH$_4$Cl (1 g/liter) (Cortecnet, Paris) as the sole carbon and nitrogen sources and 1M MgSO$_4$ (1 ml/liter), 0.1 M CaCl$_2$ (1 ml/liter), 10 mg/ml thiamine (1 ml/ liter) and 10mg/ml biotin (1 ml/liter). (Sigma-Aldrich). Induction of protein expression with IPTG and purification using histidine affinity column was performed in the same way as for the unlabeled recombinant PrP.

### Generation of transgenic mice expressing bank vole PrP I109

The bank vole PrP (BvPrP) encoding isoleucine at residue 109 was synthesized according to the BvPrP sequences (GeneBank accession numbers: AF367624.1 and EF455012.1) by mutating the residue 109 codon ATG to ATT. Transgenic vector, MoPrP.Xho was modified by replacing the XhoI with BsiWI and FseI. The coding sequence of BvPrP I109 was cloned into BsiWI and FseI sites of the modified transgenic vector and the coding sequence was confirmed by DNA sequencing. The transgene expression cassette was then released with NotI and microinjected into pronuclei of fertilized FVB/N oocytes. One transgenic line, referred to as Tg(BvPrP-I109)C594+/–, was established. The expression level of BvPrP$^C$ in the brains of Tg mice was determined by Western blotting assay using serial dilutions of brain homogenates and was compared to that of wild type FVB mice. Tg(BvPrP-I109)C594+/- mice express the same level of PrP$^C$ as the wild type FVB mice.

### *In vitro* propagation of prions by PMCA and PMSA

*Preparation of recombinant PrP-based in vitro propagation substrates* was performed as described previously for recombinant PrP-based substrates, either complemented with brain homogenates from *Prnp*$^{0/0}$ transgenic mice [9, 10, 71, 72] or with dextran sulfate sodium salt,

from *Leuconostoc spp*. with molecular weights ranging from 6,500 to 10,000 (Sigma-Aldrich) [10]. Briefly, the purified rec-PrP stored with 6 M of guanidine-HCl was diluted 1:5 in phosphate buffered saline (PBS) and dialyzed against PBS at 1:2,000 ratio for 1 h at room temperature. The dialyzed sample was centrifuged at 19,000 g for 15 min at 4°C and the supernatant used for substrate preparation. Rec-PrP concentration in the supernatant was measured (BCA protein assay kit, Thermo Scientific) and adjusted to the working concentration, which, unless otherwise indicated, was of 20 μM to reach a final concentration of 2 μM when diluted in the substrate. The protein, after dialysis and concentration adjustment was mixed with conversion buffer (CB) [71] 1:9 and dextran sulfate sodium salt, from *Leuconostoc spp*. with molecular weights ranging from 6,500 to 10,000 (Sigma-Aldrich), was added to a final concentration of 0.5% (w/v). The substrate was aliquoted and stored at -80°C until required.

**Preparation of brain PrP-based in vitro propagation substrates.** Brain PrP-based substrates from transgenic mice expressing 4-fold or 1-fold of bank vole I109 PrP [TgVole (4x) and TgVole (1x), respectively] under the control of the murine PrP promoter in a murine *Prnp^{0/0}* background which were generated and characterized in a similar way as described previously [73], were prepared also as previously described [10]. Briefly, perfused whole brains, were homogenized at 10% (w/v) in CB with protease inhibitor cocktail (Roche) in a glass potter pestle (Fisher scientific), aliquoted and stored at -80°C until required.

*Brain-PMCA* was performed based on modified versions of the PMCA described previously [74–76], to estimate the potential *in vivo* infectivity of the recombinant seeds generated prior to bioassays [10]. Using TgVole (4x) or TgVole (1x) brain homogenates as substrates, seeded at 1:10 dilution for the first 24 h round of PMCA, in a S-4000 Misonix sonicator with microplate system (Qsonica) with incubation cycles of 30 min followed by sonication pulses of 20 s at 80% power at 38°C regulated by a circulating water bath. To avoid cross-contamination, all PMCA tubes were sealed with plastic film (Parafilm) prior to introduction in the bath sonicator to prevent accidental opening. For serial rounds of PMCA, the product of the previous round was diluted 1:10 in fresh substrate and another 24 h round of PMCA performed under identical conditions. After each round of PMCA, the external surfaces of all the tubes were cleaned thoroughly with sodium hypochlorite and tubes containing different seeds or from different experiments were treated separately. 1 mm zirconium silicate beads (BioSpec Products) were included in each reaction to favor the reproducibility of the results [77]. Unseeded controls were included together with all the seeded samples subjected to brain-PMCA. Propagation was determined by PK-digestion of the PMCA products followed by Western blotting.

*recPMCA* was performed as described previously [9, 71, 72, 78] using a Q-700 Misonix sonicator with microplate system (Qsonica) with incubation cycles of 30 min, followed by sonication pulses of 15–20 s at 60–80% power at 37–39°C regulated by a circulating water bath. To determine the propagation capacity of each recombinant seed they were diluted serially 1:10 in recombinant PrP-based substrate containing recombinant bank vole I109 PrP complemented with dextran sulfate. Dilutions from $10^{-1}$ to $10^{-7}$ were subjected to a single 24 h round of recPMCA. Again, all the PMCA tubes were sealed with plastic film prior to introduction in the bath sonicator to prevent accidental opening and 1 mm zirconium silicate beads (BioSpec Products) were added to favor the reproducibility of the results [77].

*PMSA* was performed using rec-PrP derived substrates complemented with dextran sulfate. Initially the temperature was set at 38°C using a Thermomixer (Eppendorf) with internal temperature control or a shaker (Monoshake, Thermo Scientific) placed inside an incubator (Nahita) and shaking at 700 rpm set to 60 s of shaking and 5 min incubation cycles for 24 h propagation rounds unless indicated otherwise. With the same substrates and tubes used for recPMCA and the addition of 1 mm zirconium silicate beads, serial rounds at 1:10 to 1:1,000 dilutions (up to 30 rounds) and serial seed dilutions (from $10^{-1}$ to $10^{-8}$) in single 24 h rounds

were performed. After optimization, all PMSA reactions were performed at 39˚C and 1,000 rpm in a continuous mode.

## Preparation of fibrillary bank vole rec-PrP in the presence of urea

The preparation of non-infectious fibrillary bank vole rec-PrP was performed following the protocol reported previously [79]. Briefly, recombinant bank vole PrP (120 μM in 6 M Gnd-HCl) was diluted to a final concentration of 20 μM in 20 mM sodium acetate buffer (pH 5) and dialyzed against 10 mM sodium acetate (pH 5). The dialyzed protein was incubated in 1 M Gnd-HCl, 3 M urea and 150 mM of NaCl in PBS (pH7) at 37˚C and continuous shaking (600 rpm) for 72 h. Fibril formation was monitored using Thioflavin T.

## Biochemical characterization of *in vitro*- and *in vivo*-generated prion strains

*Protease K digestion*: recPMCA products based on $Prnp^{0/0}$ complemented substrates were digested by mixing the sample with N-Lauroylsarcosine sodium salt (sarkosyl) (Sigma-Aldrich) 10% in PBS 1:1 (v/v) and adding proteinase K (PK) (Roche) at 85 μg/ml for 1 h at 42˚C and shaking at 450 rpm as described previously [9]. Digestion was stopped by addition of loading buffer 1x (NuPage LDS, Invitrogen). For recPMCA and PMSA products, complemented with dextran sulfate, digestion was exactly the same but with 25 μg/ml of PK. For Blue-Safe staining, 500 μl of dextran complemented PMSA or recPMCA products were digested without any additional buffer at 25 μg/ml of PK for 1 h at 42˚C and shaking at 450 rpm. After digestion samples were immediately centrifuged at 19,000 g at 4˚C for 15 min, the supernatant was discarded and the pellet resuspended and washed with 700 μl PBS. After a further 5 min at 19,000 g at 4˚C, the supernatant was discarded and the washed pellet resuspended in 15 μl of loading buffer 1x (NuPage LDS, Invitrogen). In occasional cases other PK concentrations were used and indicated. For brain-derived samples, both obtained from diseased animals and brain-PMCA, samples were mixed 1:1 (v/v) with a digestion buffer containing 2% Nonidet P40 (Sigma-Aldrich), 2% Tween-20 (Sigma-Aldrich) and 5% sarkosyl in PBS and digested at 85 μg/ml for 1 h at 42˚C and shaking at 450 rpm. Digestion was stopped by addition of loading buffer 1x (NuPage LDS 1X, Invitrogen).

*PK-resistant PrP detection*: for total protein detection, PK-digested and concentrated samples in loading buffer were boiled for 10 min at 100˚C and loaded onto 4–12% acrylamide gels (NuPAGE Midi gel, Invitrogen Life Technologies) and subjected to electrophoresis for 1 h and 20 min (10 min at 70 V, 10 min at 110 V and 1 h at 150 V) and stained with BlueSafe (NZY-Tech) for 1 h at room temperature. For prion protein immunodetection, Western blotting was performed as described previously [78]. Briefly, PK-digested samples were boiled for 10 min and loaded on 4–12% acrylamide gels (NuPAGE Midi gel Invitrogen Life Technologies), subjected to electrophoresis for approximately 1 h and 20 min and transferred to a PVDF membrane (Trans-Blot Turbo Transfer Pack, Bio-Rad) using the Trans-Blot Turbo transfer system (Bio-Rad). After blocking non-specific antibody binding of the membranes by incubation in 5% non-fat milk powder for 1 h at room temperature, monoclonal antibodies 12B2 (1:2,500), D18 (1:5,000) or Saf83 (1:400) were added at the indicated dilutions and incubated for 1 h at room temperature, prior to washing, incubation with peroxidase-conjugated secondary anti-mouse antibody (m-IgGκ BP-HRP, Santa Cruz Biotechnology) or goat anti-human IgG (H+L, Thermo Scientific) and developed with an enhanced chemiluminescent horseradish peroxidase substrate (West Pico Plus, Thermo Scientific).

*Epitope mapping*: the characterization of misfolded recombinant PrP fragments resulting from PK digestion by epitope mapping was performed using the same protease digested

sample (25 μg/ml for 1 h at 42˚C and shaking at 450 rpm) processed for BlueSafe staining and same amount loaded in multiple lanes of 4–12% acrylamide gels. After checking that each lane contained the same amount of protein by BlueSafe staining, the gel was transferred to a PVDF membrane (Trans-Blot Turbo Transfer Pack, Bio-Rad) using the Trans-Blot Turbo transfer system (Bio-Rad). The membranes were blocked to prevent non-specific antibody binding in 5% non-fat milk powder for 1 h at room temperature, individual lanes were cut and incubated separately with the following antibodies: 12B2 (1:2,500), 9A2 (1:4,000), 7D9 (1:1,000), D18 (1:5,000), Saf83 (1:400), Sa84 (1:400) and POM19 (1:10,000).

*ESI-TOF*: for mass spectrometry-based analysis of PK-resistant fragments of L-seeded-PMSA, the pellet from 2 ml of PK-treated sample was resuspended in 70 μl of 6M Guanidine-HCl by 3 pulses of tip sonication and incubated at 37˚C for 1 h. Trifluoracetic acid was then added to a final concentration of 1%. 4 μl of the sample were injected into a micro liquid chromatography system (Eksigent Technologies nanoLC 400, SCIEX) coupled to high speed Triple TOF 6600 mass spectrometer (SCIEX) with a micro flow source. The sample was injected to a YMC-TRIART C18 trap column (YMC Technologies, Teknokroma) with a 3 mm particle size and 120 Å pore size. The loading pump delivered a wash solution of 0.1% formic acid in water at 10 μl/min. Then the sample was eluted and fed into a silica-based 150 × 0.30 mm, 3 mm particle size and 120 Å pore size Chrom XP C18 reversed phase analytical column (Eksigent, SCIEX) by a micro-pump operating at 5 μl/min and applying a gradient consisting of 0.1% formic acid in water as mobile phase A, and 0.1% formic acid in acetonitrile as mobile phase B. Peptides were separated using a 40 min gradient ranging from 2% to 90% mobile phase B (mobile phase A: 2% acetonitrile, 0.1% formic acid; mobile phase B: 100% acetonitrile, 0.1% formic acid). TOF MS data were acquired in a TripleTOF 6600 System (SCIEX, Foster City, CA) using the following parameters; source and interface conditions were ion spray voltage floating (ISVF) 5500 V, curtain gas (CUR) 25, collision energy (CE) 10 and ion source gas 1 (GS1) 25. Finally, the protein was reconstructed using the Bio tool integrated algorithm. Data were deconvolved using a m/z range of 350–1400.

*Evaluation of size distribution of aggregates by sedimentation velocity analysis in continuous sucrose gradient*: Based on experiments performed previously [55], adapted to the recombinant samples devoid of any material but 5% (w/v) dextran sulfate, 1% (w/v) Triton-X-100 and 0.15 M NaCl. A wide range continuous sucrose gradient was prepared from 10% (top) to 80% (bottom) (w/v) expecting large aggregates and using a total volume of 12.5 ml in order to achieve a good resolution in case of high heterogeneity of aggregate size. The gradient was prepared by diffusion of a discontinuous gradient that included 2.5 ml layers of 80-60-40-20 and 10% (w/v) sucrose solutions with 0.2% of sarkosyl to favor correct partition of weakly interacting PrP species. The discontinuous gradient, prepared on ultra-clear centrifuge tubes (14x89 mm, Beckman Coulter) sealed with parafilm, were carefully rotated to horizontal position and incubated overnight at 4˚C before loading the sample. Immediately before ultracentrifugation, 2.5 ml from the top were replaced by 2.5 ml of either PK-digested (85 μg/ml for 1 h at 42˚C) or undigested L-seeded-PMSA and centrifuged at 40,000 rpm (approximately 200,000 *g*) for 18 h in a swinging-bucket SW-41 rotor using an Optima ultracentrifuge (Beckman Coulter). After centrifugation, 500 μl fractions were manually collected from top to bottom and pellet also resuspended in 500 μl and processed as follows for western blotting: 4 μl of each fraction of the undigested sample was diluted in 16 μl of PBS and 10 μl of NuPage 4X loading buffer, loading 15 μl from this mix in the gel. 200 μl of each PK-digested sample fraction were precipitated with methanol, resuspended in 30 μl of 1:3 PBS and NuPage 4X buffer from which 10 μl were loaded. Thus, approximately 30 times more sample was loaded for PK-digested sample with respect to undigested sample. Western blot was performed as described in previous sections and developed with Saf83 mAb (1:400).

## Biological characterization of *in vitro* generated prion strains

*Preparation of in vivo and in vitro derived inocula*: 10% brain homogenates from vole-adapted chronic wasting disease (CWD) were diluted $10^{-1}$ in sterile PBS prior to intracerebral inoculation into TgVole (1x) and bank voles 109I (Bv109I). Equally, PMCA and PMSA products were diluted $10^{-1}$ in sterile PBS prior to intracerebral inoculation in the same models. The protease-resistant rec-PrP amount, estimated by Western blotting, was comparable in all samples. The inocula for the second passage were prepared, as 10% (w/v) brain homogenates in PBS form the first passage animals.

*Animal inoculations*: groups of eight-week-old bank voles (Bv109I) were inoculated intracerebrally with 20 µl of homogenate into the left cerebral hemisphere using a sterile disposable 27-gauge hypodermic needle while under ketamine anesthesia (ketamine 0.1 µg/g). Groups of 5 to 10 TgVole (1x) animals were inoculated in the same way but using tribromoethanol (Avertin) (0.5 mg/g) as anesthetic. The animals were examined twice a week until neurological clinical signs appeared, after which they were examined daily. Clinically affected animals were culled at the terminal stage of the disease, but before neurological impairment compromised their welfare, by exposure to a rising concentration of carbon dioxide. Survival time was calculated as the interval between inoculation and culling or death. *Post-mortem*, the brain was removed and divided sagitally to be stored at -80˚C and fixed in formalin.

*Neuropathology*: histology and immunohistochemistry were performed on formalin-fixed tissues as described previously [80]. Briefly, brains were trimmed at standard coronal levels, dehydrated through graded alcohols, embedded in paraffin wax, sectioned (6 µm) and stained with hematoxylin and eosin. Immunohistochemistry for PrP was performed using the 6C2 mAb (1:300) as described previously [81].

## Preparation of recombinant prions for structural studies

Large volumes of PMSA products for structural studies, such as solid state Nuclear Resonance (ssNMR), were generated in 5 ml tubes (Eppendorf) at optimized PMSA conditions and the resulting products were digested using 25 µg/ml of PK for 1 h at 42˚C and centrifuged immediately at 19,000 g for 1 h at 4˚C. The pellets were washed with PBS and centrifuged again at 19,000 g for 15 min at 4˚C. The supernatants were discarded and the pellet resuspended in water.

## Solid state Nuclear Magnetic Resonance

For MAS NMR measurements, the sample was sealed hermetically in a custom-built unbreakable polyformaldehyde rotor insert designed to fit into 3.2 mm Varian rotors. The $^{13}C$ NMR signal of this material at 89.1 ppm does not interfere with the protein spectra [82]. Solid-state NMR spectra were recorded using a 3.2 mm MAS probe head in double resonance mode ($^{1}H, ^{13}C$) at a static magnetic field of 14.1 T (Varian) and a MAS frequency of 11 kHz ± 3 Hz, at a nominal temperature of 0˚C (temperature of the cooling gas). The $^{1}H$ 90˚ pulse length was 3 µs, the $^{13}C$ 90˚ pulse length 6.5 µs. The Proton-driven spin-diffusion (PDSD) experiment was recorded with a longitudinal mixing time of 50 ms, during indirect evolution and detection SPINAL-64 proton decoupling [83] at an rf field strength of 83 kHz. 196 complex points were recorded in the indirect dimension, corresponding to a maximum $t_1$ acquisition time of 6 ms. In total five spectra with 128, 176 and 3·320 scans were recorded and summated, such that the total number of scans was 1,264. The 2D spectrum was processed using the squared cosine window functions in both dimensions and referenced to DSS by using adamantane as an external reference. A control INEPT spectrum, which exclusively displays signals of highly mobile protein regions, was recorded with 400 scans at a nominal temperature of 15˚C, but no signals were visible [47].

## Ethics statement

TgVole mice were obtained from the breeding colony at CIC bioGUNE (Spain) and were inoculated at the University of Santiago de Compostela. All experiments involving animals in Spain adhered to the guidelines included in the Spanish law "Real Decreto 1201/2005 de 10 de Octubre" on protection of animals used for experimentation and other scientific purposes, which is based on the European Directive 86/609/EEC on Laboratory Animal Protection. The project was approved by the Ethical Committees on Animal Welfare (project codes assigned by the Ethical Committee P-CBG-CBBA-0314 and 15005/16/006) and performed under their supervision. Bank voles were obtained from the breeding colony at the Istituto Superiore di Sanità (ISS), Italy. Experiments involving animals followed the "Principles of laboratory animal care" (NIH publication No. 86–23, revised 1985) as well as the guidelines contained in the Italian Legislative Decree 116/92, which is based on the European Directive 86/609/EEC on Laboratory Animal Protection, and then in the Legislative Decree 26/2014, which transposed the European Directive 2010/63/UE on Laboratory Animal Protection. The research protocol was performed under the supervision of the Service for Biotechnology and Animal Welfare of the ISS and was approved by the Italian Ministry of Health (decree number 84/12.B).

## Supporting information

**S1 Fig. Shaking system and *ad hoc* designed racks for different tube sizes.** Photograph on the left shows the rack designed with 96 wells for 0.2 ml PCR tubes on the shaker used for PMSA reactions. The photograph on the right shows the rack with 6 horizontal cells for 5 ml tubes suitable for the same shaker. Both racks were designed *ad hoc* and 3D-printed (I+3D). Racks for any desired tube can be designed and printed making PMSA a highly versatile method for misfolded protein production.
(PDF)

**S2 Fig. PrP expression levels in TgVole (1x) animals compared to PrP expression levels of bank vole I109 by Western blot.** 10% brain homogenates from TgVole (1x) (I109I) mouse and bank vole 109I were diluted 1:16, 1:32, 1:64 and 1:128 and analyzed by Western blot using monoclonal antibody D18 (1:5,000). The PrP expression levels of TgVole (1x) were equal to PrP$^C$ levels in bank vole brain based on signal intensity. No significant differences were observed in the electrophoretic migration patterns. Mw: Molecular weight.
(PDF)

**S3 Fig. Sedimentation velocity analysis of L-seeded-PMSA by ultracentrifugation through a continuous sucrose gradient.** Proteinase K-digested and undigested L-seeded-PMSA were subjected to ultracentrifugation at ~200,000 g through a continuous sucrose gradient ranging from 10 to 80% (w/v) with 0.2% sarkosyl to determine the size distribution of total PrP and PrPres assemblies by Western blot. The equivalent of 2 **μ**l of each fraction of the gradient were loaded for the undigested sample, while for PK-digested sample, fractions were precipitated with methanol loading the equivalent to 60 **μ**l, thus PK-digested gel shows 30-fold higher sample amount than the gel of undigested fractions. Briefly, the sedimentation pattern shows total absence of monomeric PrP in the top fractions and a continuum in the aggregate size from fractions containing around 30% to 80% sucrose for the untreated sample and from around 40% to 80% sucrose for the PK-digested sample, suggesting predominance of large PK-resistant assemblies in the sample. This indicates the presence of a heterogeneous size assemblies but does not reveal clearly different majoritarian populations that could point towards the existence of distinct assembly types regarding tertiary structure. Developed with Saf83 mAb

(1:400). PK: Proteinase K, MW; molecular weight marker.
(PDF)

**S1 Table. L-seeded-PMSA protease K-resistant fragments identified by ESI-TOF.**
(PDF)

## Acknowledgments

The authors would like to thank the following for their support: the IKERBasque Foundation, vivarium and maintenance from CIC bioGUNE and Patricia Piñeiro for technical support; David Eguiarte for designing and implementing the first 3D-printed rack; B. Esters and H. Pei for assistance with protein expression; Dr. H. Müller and Dr. Piechatzek for fruitful discussions. Access to the Jülich-Düsseldorf Biomolecular NMR Center is gratefully acknowledged. Dr. Mark P. Dagleish (Moredun Research Institute) for useful discussion and advice. The authors would also like to acknowledge Adriano Aguzzi for kindly providing POM19 mAb.

## Author Contributions

**Conceptualization:** Hasier Eraña, Natalia Fernández-Borges, Jesús R. Requena, Joaquín Castilla.

**Funding acquisition:** Jesús R. Requena, Joaquín Castilla.

**Investigation:** Hasier Eraña, Jorge M. Charco, Michele A. Di Bari, Carlos M. Díaz-Domínguez, Rafael López-Moreno, Enric Vidal, Ezequiel González-Miranda, Miguel A. Pérez-Castro, Sandra García-Martínez, Susana Bravo, Natalia Fernández-Borges, Mariví Geijo, Claudia D'Agostino, Jifeng Bian, Anna König, Boran Uluca-Yazgi, Raimon Sabate, Vadim Khaychuk, Ilaria Vanni, Glenn C. Telling, Henrike Heise, Romolo Nonno, Jesús R. Requena, Joaquín Castilla.

**Methodology:** Jorge M. Charco, Michele A. Di Bari, Carlos M. Díaz-Domínguez, Rafael López-Moreno, Enric Vidal, Ezequiel González-Miranda, Miguel A. Pérez-Castro, Sandra García-Martínez, Susana Bravo, Mariví Geijo, Claudia D'Agostino, Joseba Garrido, Jifeng Bian, Anna König, Boran Uluca-Yazgi, Raimon Sabate, Vadim Khaychuk, Ilaria Vanni, Glenn C. Telling, Romolo Nonno, Jesús R. Requena.

**Project administration:** Joaquín Castilla.

**Supervision:** Mariví Geijo, Joseba Garrido, Glenn C. Telling, Henrike Heise, Jesús R. Requena, Joaquín Castilla.

**Writing – original draft:** Hasier Eraña, Henrike Heise, Jesús R. Requena, Joaquín Castilla.

**Writing – review & editing:** Hasier Eraña, Jorge M. Charco, Michele A. Di Bari, Carlos M. Díaz-Domínguez, Rafael López-Moreno, Enric Vidal, Ezequiel González-Miranda, Miguel A. Pérez-Castro, Sandra García-Martínez, Natalia Fernández-Borges, Joseba Garrido, Anna König, Glenn C. Telling, Henrike Heise, Romolo Nonno, Jesús R. Requena, Joaquín Castilla.

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
