## [Decision Letter · Decision Letter 0]

6 Aug 2019

Dear Dr. Castilla,

Thank you very much for submitting your manuscript "Development of a new largely scalable in vitro prion propagation method for the production of infectious recombinant prions for high resolution structural studies" (PPATHOGENS-D-19-01177) for review by PLOS Pathogens. Your manuscript was fully evaluated at the editorial level and by independent peer reviewers. The reviewers appreciated the attention to an important problem, but raised some substantial concerns about the manuscript as it currently stands. These issues must be addressed before we would be willing to consider a revised version of your study. We cannot, of course, promise publication at that time.

We therefore ask you to modify the manuscript according to the review recommendations before we can consider your manuscript for acceptance. Your revisions should address the specific points made by each reviewer.

(1) A letter containing a detailed list of your responses to the review comments and a description of the changes you have made in the manuscript. Please note while forming your response, if your article is accepted, you may have the opportunity to make the peer review history publicly available. The record will include editor decision letters (with reviews) and your responses to reviewer comments. If eligible, we will contact you to opt in or out.

(2) Two versions of the manuscript: one with either highlights or tracked changes denoting where the text has been changed; the other a clean version (uploaded as the manuscript file).

Additionally, to enhance the reproducibility of your results, PLOS recommends that you deposit your laboratory protocols in protocols.io, where a protocol can be assigned its own identifier (DOI) such that it can be cited independently in the future. For instructions see http://journals.plos.org/plospathogens/s/submission-guidelines#loc-materials-and-methods

We hope to receive your revised manuscript within 60 days. If you anticipate any delay in its return, we ask that you let us know the expected resubmission date by replying to this email. Revised manuscripts received beyond 60 days may require evaluation and peer review similar to that applied to newly submitted manuscripts.

Sincerely,

David Westaway

Section Editor

PLOS Pathogens

Kasturi Haldar

Editor-in-Chief

PLOS Pathogens

orcid.org/0000-0001-5065-158X

Grant McFadden

Editor-in-Chief

PLOS Pathogens

orcid.org/0000-0002-2556-3526

Reviewer's Responses to Questions

**Part I - Summary**

Reviewer #1: The work by Eraña et al. describes a novel methodology for obtaining large quantities of fairly defined prions for detailed structural studies. The work is well-described and executed.

Reviewer #2: In this submitted manuscript, Erana et al adapted the protein misfolding cyclic amplification (PMCA) assay to dramatically improve the amount of pathological prion protein produced. This new method termed protein misfolding shaking amplification assay (PMSA) is based o the use of shaking instead of sonication.

The authors nicely show the quasi-equivalence between PMCA and PMSA in their capacity to generate bank vole recombinant infectious prions. They provide a pilot solid state RMN study suggesting that the amount of PMSA infectious PrPSc produced is sufficient.

**Part II – Major Issues: Key Experiments Required for Acceptance**

Reviewer #1: I would recommend to increase the number of animals in wild-type Bank Vole experiment inoculated with L-seeded-PMSA (1).

Reviewer #2: I have a major concern which I think must be addressed with RMN studies in mind. The authors show that PMSA generates a major type of PrPSc with regard to their strain properties. However, they did not address the heterogeneity of the PrPSc assemblies produced by PMSA with respect to their quaternary structures. Those are likely to greatly vary, as previously shown by other authors (Silveira et al, Nature 2005; Tixador et al, PloS Pathogens 2010, etc). This quaternary structural heterogeneity might make RMN studies and their interpretation (which assembly is finally studied and what is its relevance to the disease phenotype?) much more complex. To address this point, experiments such as sedimentation velocity or size exclusion chromatography should provide elements on the dispersity of the assemblies produced by PMSA.

**Part III – Minor Issues: Editorial and Data Presentation Modifications**

Reviewer #1: In Table 1 the authors should specify (1) and (2)

Why do authors find differences between the incubation times in TgVole (1X) (Table 1) and wild-type Bank Vole (Table 2) using L-seeded-PMSA (1). It should be very similar if the expression level of PrP in TgVole (1X) is the same as in wild-type animals or if the infection units inoculated in both experiments are the same. This point should be discussed.

Reviewer #2: As it is, the manuscript is very well written and clear.

PLOS authors have the option to publish the peer review history of their article (what does this mean?). If published, this will include your full peer review and any attached files.

Reviewer #1: No

Reviewer #2: No

---

## [Decision Letter · Decision Letter 1]

1 Oct 2019

Dear Dr. Castilla,

We are pleased to inform that your manuscript, "Development of a new largely scalable in vitro prion propagation method for the production of infectious recombinant prions for high resolution structural studies", has been editorially accepted for publication at PLOS Pathogens. 

Before your manuscript can be formally accepted and sent to production, you will need to complete our formatting changes, which you will receive by email within a week. Please note that your manuscript will not be scheduled for publication until you have made the required changes.

IMPORTANT NOTES

(1) Please note, once your paper is accepted, an uncorrected proof of your manuscript will be published online ahead of the final version, unless you’ve already opted out via the online submission form. If, for any reason, you do not want an earlier version of your manuscript published online or are unsure if you have already indicated as such, please let the journal staff know immediately at plospathogens@plos.org.

(2) Copyediting and Proofreading: The corresponding author will receive a typeset proof for review, to ensure errors have not been introduced during production. Please review the PDF proof of your manuscript carefully, as this is the last chance to correct any errors. Please note that major changes, or those which affect the scientific understanding of the work, will likely cause delays to the publication date of your manuscript. 

(3) Appropriate Figure Files: Please remove all name and figure # text from your figure files. Please also take this time to check that your figures are of high resolution, which will improve the readbility of your figures and help expedite your manuscript's publication. Please note that figures must have been originally created at 300dpi or higher. Do not manually increase the resolution of your files. For instructions on how to properly obtain high quality images, please review our Figure Guidelines, with examples at: http://journals.plos.org/plospathogens/s/figures.

(4) Striking Image: Please upload a striking still image to accompany your article if one is available (you can include a new image or an existing one from within your manuscript). Should your paper be accepted, this image will be considered for our monthly issue image and may also appear on our website to feature your article. Please upload this as a separate file, selecting "striking image" as the file type upon upload. Please also include a separate "Other" file with a caption, including credits and any potential copyright information. Please do not include the caption in the main article file. If your image is from someone other than yourself, please ensure that the artist has read and agreed to the terms and conditions of the Creative Commons Attribution License at http://journals.plos.org/plospathogens/s/content-license. Please note that PLOS cannot publish copyrighted images.

(5) Press Release or Related Media: If your institution or institutions have a press office, please notify them about your upcoming paper at this point, to enable them to help maximize its impact. If they will be preparing press materials for this manuscript, please inform our press team in advance at plospathogens@plos.org as soon as possible. We ask that you contact us within one week to plan ahead of our fast Production schedule. If you need to know your paper's publication date for related media purposes, you must coordinate with our press team, and your manuscript will remain under a strict press embargo until the publication date and time. This means an early version of your manuscript will not be published ahead of your final version. 

(6)  PLOS requires an ORCID iD for all corresponding authors on papers submitted after December 6th, 2016. Please ensure that you have an ORCID iD and that it is validated in Editorial Manager.  To do this, go to ‘Update my Information’ (in the upper left-hand corner of the main menu), and click on the Fetch/Validate link next to the ORCID field.  This will take you to the ORCID site and allow you to create a new iD or authenticate a pre-existing iD in Editorial Manager

(7) Update your Profile Information: Now that your manuscript has been provisionally accepted, please log into Editorial Manager and update your profile, if needed. Go to https://www.editorialmanager.com/ppathogens, log in, and click on the "Update My Information" link at the top of the page. Please update your user information to ensure an efficient production and billing process. 

(8) LaTeX users only: Our staff will ask you to upload a TEX file in addition to the PDF before the paper can be sent to typesetting, so please carefully review our Latex Guidelines http://journals.plos.org/plospathogens/s/latex in the meantime.

(9) If you have associated protocols in protocols.io, please ensure that you make them public before publication to guarantee immediate access to the methodological details.

Best regards,

David Westaway

Section Editor

PLOS Pathogens

Kasturi Haldar

Editor-in-Chief

PLOS Pathogens

orcid.org/0000-0001-5065-158X

Grant McFadden

Editor-in-Chief

PLOS Pathogens

orcid.org/0000-0002-2556-3526

Reviewer Comments (if any, and for reference):

Reviewer's Responses to Questions

Part I - Summary

Reviewer #2: The authors have addressed my concerns satisfactorily. Congratulations for this nice piece of work.

Part II – Major Issues: Key Experiments Required for Acceptance

Please use this section to detail the key new experiments or modifications of existing experiments that should be 

absolutely

 required to validate study conclusions.

Reviewer #2: none

Part III – Minor Issues: Editorial and Data Presentation Modifications

Reviewer #2: none

PLOS authors have the option to publish the peer review history of their article (what does this mean?). If published, this will include your full peer review and any attached files.

Do you want your identity to be public for this peer review?

 For information about this choice, including consent withdrawal, please see our Privacy Policy.

Reviewer #2: No

---

## [Editor Report · Acceptance letter]

16 Oct 2019

Dear Dr. Castilla,

We are delighted to inform you that your manuscript, "Development of a new largely scalable in vitro prion propagation method for the production of infectious recombinant prions for high resolution structural studies," has been formally accepted for publication in PLOS Pathogens.

The corresponding author will soon be receiving a typeset proof for review, to ensure errors have not been introduced during production. Please review the PDF proof of your manuscript carefully, as this is the last chance to correct any scientific or type-setting errors. Please note that major changes, or those which affect the scientific understanding of the work, will likely cause delays to the publication date of your manuscript. Note: Proofs for Front Matter articles (Pearls, Reviews, Opinions, etc…) are generated on a different schedule and may not be made available as quickly.

Best regards,

Kasturi Haldar

Editor-in-Chief

PLOS Pathogens

orcid.org/0000-0001-5065-158X

Grant McFadden

Editor-in-Chief

PLOS Pathogens

orcid.org/0000-0002-2556-3526